# Transposable element-initiated enhancer-like elements generate the subgenome-biased spike specificity of polyploid wheat

Yilin Xie[1,2,3,11], Songbei Ying [1,11], Zijuan Li[2,3,11], Yu'e Zhang [3,4,11], Jiafu Zhu[5,11], Jinyu Zhang [1,2,3], Meiyue Wang[1], Huishan Diao[1], Haoyu Wang[2,6], Yuyun Zhang[1,2,3], Luhuan Ye[2,3], Yili Zhuang[2,3], Fei Zhao[2,3], Wan Teng[3,4], Wenli Zhang [7], Yiping Tong[3,4], Jungnam Cho [8] ✉, Zhicheng Dong [5] ✉, Yongbiao Xue [3,4,9,10] ✉ & Yijing Zhang [1] ✉

Transposable elements (TEs) comprise ~85% of the common wheat genome, which are highly diverse among subgenomes, possibly contribute to polyploid plasticity, but the causality is only assumed. Here, by integrating data from gene expression cap analysis and epigenome profiling via hidden Markov model in common wheat, we detect a large proportion of enhancer-like elements (ELEs) derived from TEs producing nascent noncoding transcripts, namely ELE-RNAs, which are well indicative of the regulatory activity of ELEs. Quantifying ELE-RNA transcriptome across typical developmental stages reveals that TE-initiated ELE-RNAs are mainly from RLG_famc7.3 specifically expanded in subgenome A. Acquisition of spike-specific transcription factor binding likely confers spike-specific expression of RLG_famc7.3-initiated ELE-RNAs. Knockdown of RLG_famc7.3-initiated ELE-RNAs resulted in global downregulation of spike-specific genes and abnormal spike development. These findings link TE expansion to regulatory specificity and polyploid developmental plasticity, highlighting the functional impact of TE-driven regulatory innovation on polyploid evolution.

The rise of hexaploid wheat in the Fertile Crescent about 10,000 years ago revolutionized the way modern humans live[1–3]. Common wheat (*Triticum aestivum*, AABBDD) genome consists of three sets of diploid genomes adapted to different environments and dominates global wheat production[4,5]. Subgenomic diversity is one major factor contributing to the success of common wheat[3,6]. The three subgenomes, particularly the intergenic regions, were highly divergent due to multiple rounds of transposable element (TE) expansions[5]. TEs make up

[1]State Key Laboratory of Genetic Engineering, Collaborative Innovation Center of Genetics and Development, Department of Biochemistry, Institute of Plant Biology, School of Life Sciences, Fudan University, Shanghai 200438, China. [2]National Key Laboratory of Plant Molecular Genetics, CAS Center for Excellence in Molecular Plant Sciences, Shanghai Institute of Plant Physiology and Ecology, Shanghai Institutes for Biological Sciences, Chinese Academy of Sciences, 300 Fenglin Road, Shanghai 200032, China. [3]University of the Chinese Academy of Sciences, Beijing 100049, China. [4]State Key Laboratory of Plant Cell and Chromosome Engineering, Institute of Genetics and Developmental Biology, and The Innovation Academy of Seed Design, Chinese Academy of Sciences, Beijing 100101, China. [5]Guangdong Provincial Key Laboratory of Plant Adaptation and Molecular Design, Guangzhou Key Laboratory of Crop Gene Editing, Innovative Center of Molecular Genetics and Evolution, School of Life Sciences, Guangzhou University, Guangzhou 510006, China. [6]Henan University, School of Life Science, Kaifeng, Henan 457000, China. [7]State Key Laboratory for Crop Genetics and Germplasm Enhancement, CIC-MCP, Nanjing Agricultural University, No.1 Weigang, Nanjing, Jiangsu 210095, China. [8]Department of Biosciences, Durham University, Durham DH1 3LE, United Kingdom. [9]Beijing Institute of Genomics, Chinese Academy of Sciences, and China National Centre for Bioinformation, Beijing 100101, China. [10]Jiangsu Co-Innovation Center for Modern Production Technology of Grain Crops, Yangzhou University, Yangzhou 225009, China. [11]These authors contributed equally: Yilin Xie, Songbei Ying, Zijuan Li, Yu'e Zhang, Jiafu Zhu. ✉e-mail: jungnam.cho@durham.ac.uk; zc_dong@gzhu.edu.cn; ybxue@genetics.ac.cn; zhangyijing@fudan.edu.cn

around 85% of the 16 Gb common wheat genome, which contribute a substantial amount of regulatory elements (REs) and are potentially associated with co-option of TE and the host genome[7]. However, a causal relationship between TE-embedded REs and polyploid plasticity has so far been missing. Therefore, it is unknown if the contribution of TEs to REs has functional relevance or due to genetic drift. Elucidating this issue is important for understanding the impact of TE turnovers across diploid progenitors on the polyploid plasticity during evolution, and for crop improvement by targeting REs without altering the coding sequences.

Common wheat TEs preferentially contribute to distal REs, some of which have enhancer activity[7]. Consistently, abundant chromatin loops were detected by Hi-C technology in common wheat, indicating abundant distal regulations[8,9]. However, inferring the function of distal REs is challenged by their ability to act on target genes over long and variable distances, as well as the propensity of individual enhancers to regulate multiple genes. Additional challenges are that RE activity is dynamic and restricted to specific tissue or responsive to specific environmental cues. Furthermore, functional REs have no discernible sequence signature. Therefore, new approaches are needed to functionally dissect the distal RE. Recent human studies reported prevalent transcription of noncoding transcripts (ncRNAs) from polymerase II-initiated enhancers, generating enhancer RNAs (eRNAs), some of which play diverse roles in regulating cellular functions[10–16]. Despite it remains unclear whether eRNA functions are generalizable, given that eRNA transcription occurs very early in gene transcription process, and their activities correspond well to the dynamic tissue and lineage specificity, eRNA serves as a good marker of cell state and function. Recent report in common wheat seedlings detected thousands of nascent noncoding transcripts produced by ELEs, which were much more likely to be functional than candidate REs that were not transcribed[17]. Profiling the transcriptional specificity of these ELE-RNAs is a promising strategy for deducing the regulatory functions of ELEs.

The majority of eRNAs are generally unstable, which are not readily detectable in steady-state RNA-sequencing data. Their annotation depends on sequencing nascent RNA using approaches including global run-on sequencing (GRO-seq) and cap analysis of gene expression (CAGE), the latter of which yields TSS information at nucleotide resolution, and was shown to be highly accurate for detecting 5′ ends of both mRNAs[18–20] and eRNA[10] in mammal studies. In plants, high-quality TSS was detected using CAGE and relevant technologies in Arabidopsis[21,22], maize[23] and cotton[24], mostly focusing on coding genes. The large intergenic region of common wheat harbors abundant noncoding nascent transcripts from ELEs probed by GRO-seq[17].

In this work, via hidden Markov model integrating tissue-specific maps of CAGE, epigenomic and transcriptomic profiles, we detect active nascent ncRNAs transcribed from 11,452 ELEs. We further reveal the impact of subgenome-specific TE expansion on subgenome heterogenic transcription of genes and ELEs, thereby regulating subgenome-biased developmental specificity. These findings detect TE-embedded ELEs as the anchor mediating reciprocal adaptation between transposable element and the host genome, contributing to polyploid developmental specificity and plasticity.

## Results

### Active nascent RNA transcription in common wheat transposable elements

Transposable elements (TEs) in the common wheat genome are largely repressed by hyper DNA methylation (Fig. 1a). However, a closer examination revealed local hypo-methylated TE loci comprising open chromatin surrounded by active histone marks indicative of active regulatory elements (REs) (Fig. 1b), consistent with recent report defected prevalent bindings of transcription factors inside TEs in wheat[7]. However, the function of these REs in development are difficult

to be predicted given that the majority of TE-embedded REs are distant from genes (enhancer-like elements, ELEs) and potentially influence multiple targets in a tissue-specific manner. Recent human studies demonstrated the prevalent and early production of noncoding transcripts from active enhancers, which are excellent markers for directly predicting enhancer functions in developmental specificity[10–16].

We detected apparent enrichment of nascent noncoding transcripts previously generated in seedlings (data from[17]) in these TE-embedded ELEs (Fig. 1c). In order to profile the dynamic transcriptome of ELE-RNAs, the genome-wide gene and RE TSSs were mapped on the basis of CAGE sequencing, bisulfite sequencing (BS-seq) and chromatin immunoprecipitation sequencing (ChIP-seq) of three major histone marks (H3K4me3, H3K27me3, and H3K9ac) in four Chinese Spring tissues (seedling, root, embryo, and spike) (Fig. 1d–e and Supplementary Data 1). A total of 117,757 CAGE-tag clusters (i.e., regions enriched with TSS reads) were identified and annotated via the hidden Markov model integrating reference gene models, RNA-seq data, and epigenetic data (see Methods) (Fig. 1f, Supplementary Fig. 1a–e), resulting in 50,362 high-confidence mRNA TSSs, 44,930 low-confidence mRNA TSSs, and 11,452 ELE TSSs (Supplementary Fig. 1f and Supplementary Data 2). The comprehensive TSS annotation of coding genes enriched the annotation of alternative TSSs, upstream open reading frames and sharp or broad types of TSS, shedding new light on the mechanism of transcriptional regulation and alternative protein products (Supplementary Note 1 and Supplementary Figs. 2–8). Here, we focused on the detection and characterization of TSSs of transcripts from ELEs described as follows.

Considerable intergenic transcription was detected, accounting 16-25% of the TSS clusters (Fig. 1g). These intergenic TSS clusters were highly diverse among subgenomes (Supplementary Fig. 9). Compared with the total RNA-seq signals for these intergenic TSSs, the CAGE signals were more abundant because they included the unstable nascent RNAs that were undetectable during the RNA-seq analysis (Fig. 1h, e)[10,25]. 60% (11,452) of the intergenic TSS clusters overlapped with H3K9ac and/or H3K4me3 (i.e., typical enhancer markers in plants[26–28]) (Supplementary Fig. 1e and Supplementary Data 2); these are referred to as ELE-RNA TSSs. The ELE-RNA TSSs clearly defined the boundaries of H3K4me3 and H3K9ac (Fig. 1i), confirming that these transcripts originate from the edges of ELEs. ELE-RNA TSS regions were highly enriched with nascent transcript signals detected by other strategies including plant native elongating transcript sequencing and global run-on sequencing previously[17] (Supplementary Fig. 10). The genetic and epigenetic signatures that distinguished ELE-RNA TSSs from gene TSSs were comprehensively profiled (Supplementary Note 2, Supplementary Figs. 11–13 and Supplementary Data 3). The transcription factor binding sites differentially enriched for ELE-RNA and gene TSS were profiled. ELE-RNA and gene TSS sequences could be apparently distinguished by the machine learning approach (AUC value > 0.86), representing a high average of true positive rates over all possible values of the false positive rate (Supplementary Note 2 and Supplementary Figs. 11–13). The comprehensive ELE-RNA TSS atlas and relevant epigenetic signatures are useful resources for predicting the regulatory specificity of ELEs in common wheat.

### CAGE identifies subgenome-partitioned tissue-specific transcription from TE-embedded ELEs

We further examined the ELE-RNAs expressed in specific tissues (Fig. 2a and Supplementary Data 4). ELE-RNAs are largely correlated with nearby gene expression levels (Supplementary Fig. 14), based on which, we developed a statistical strategy to define ELE-RNA targets by integrating information from gene proximity and expression correlation (please refer to the Methods), resulting in the definition of a putative target list with high degree of consistent tissue specificity with the corresponding ELE-RNAs (Fig. 2b, c). Compared with the ubiquitously expressed ELE-RNAs, the sequences of tissue-specifically

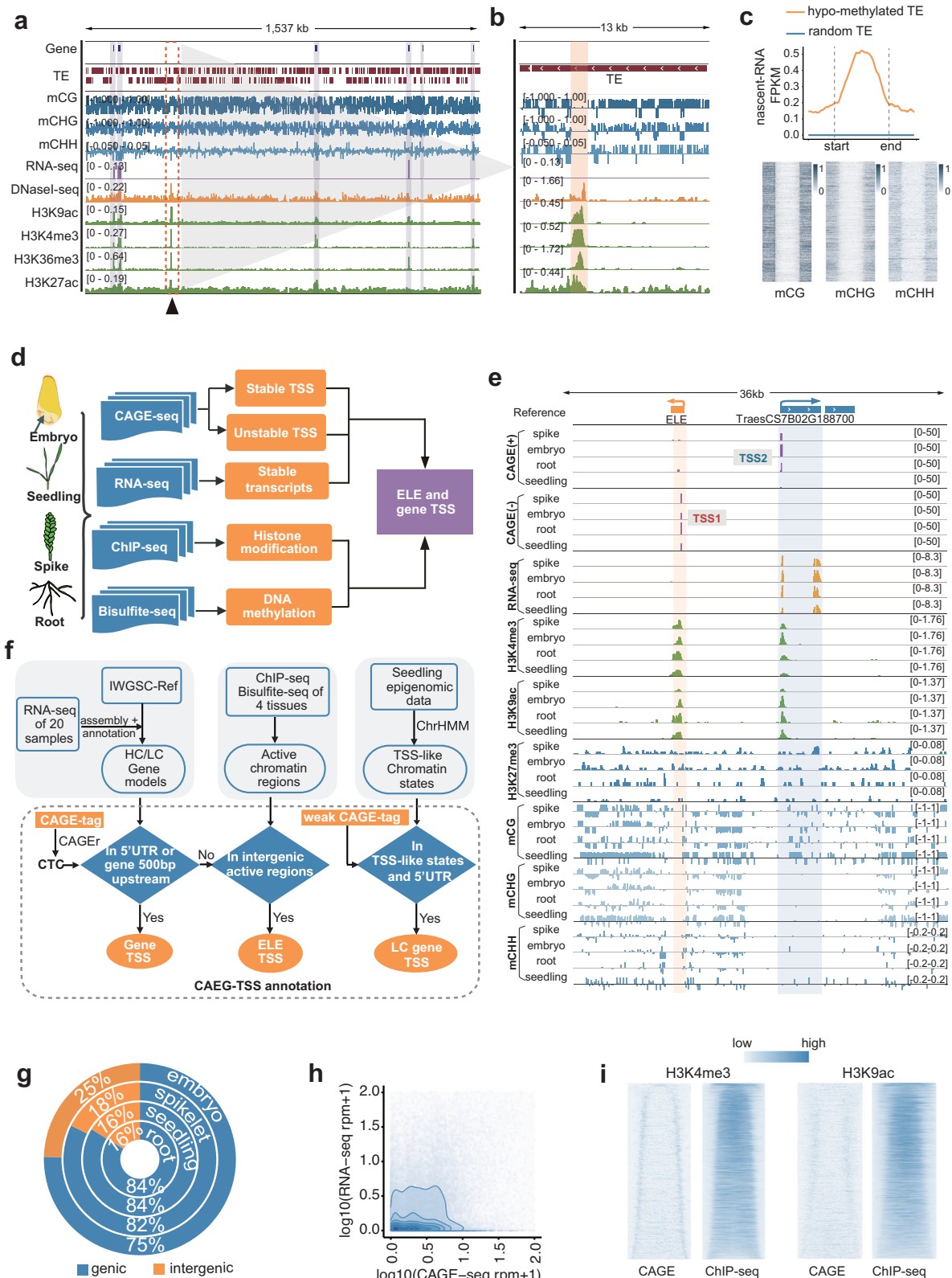

expressed ELE-RNAs were less conserved across subgenomes (Fig. 2d), indicating a subset of tissue-specific functions of ELEs evolved independently across subgenomes. Subgenome A-specific ELE-RNAs were preferentially expressed in spikes, whereas ELE-RNAs specific to subgenomes B and D were highly expressed in seedlings (Fig. 2e), implying that subgenome-specific ELE-RNAs may differentially contribute to developmental specificity.

Given that heterogeneous TE expansion is the major cause of subgenome diversity, we next focused on the TE origin of subgenome-diversified ELE-RNAs. Approximately 20% of the ELE-RNA TSSs are embedded in TEs, whereas only approximately 5% of gene TSS clusters overlapped with TEs (Fig. 2f), thus, TEs preferentially promote innovation of gene distal regulation during evolution. The TEs are generally repressed via epigenetic mechanisms (i.e., DNA methylation)[29]. We

**Fig. 1 | De novo identification of TSSs related to common wheat development.**
**a** Genomic tracks illustrating the global repression of TEs by DNA methylation.
Genes are marked with gray rectangles. The region in the orange dashed box is
enlarged in **b**. **b** Genomic tracks illustrating a local hypo-methylated TE locus with
open chromatin and active histone modifications (expanded view from Fig. 1a).
**c** Top: average ELE-RNA expression profiles determined by the nascent RNA-
sequencing analysis surrounding hypo-methylated and random TEs. FPKM, frag-
ments per kilobase of transcript per million mapped fragments. Bottom: heatmaps
of CG, CHG, and CHH DNA methylation rates around each hypo-methylated TE
locus that overlaps with an ELE-RNA. **d** Workflow of the experimental design. **e.**
Genomic tracks illustrating the CAGE signals in the TSSs of the genes and ELEs in
four tissues. CAGE(+/−) represents the positive/negative strand for the CAGE-seq
analysis. **f** Workflow of the genome-wide TSS annotation based on the integration
of CAGE-seq, transcriptome, and epigenome data. CAGE-seq data were generated
from embryo, seedling, spike and root. The CAGE-TSS is defined as a region with an
enriched CAGE signal detected by CAGEr. CAGE-TSSs located at the 5′-end of

annotated genes, or de novo assembled transcripts with coding potential are
defined as gene TSSs, whereas intergenic noncoding CAGE-TSSs overlapped with
active epigenetic markers indicative of enhancer activity, including enrichment of
H3K4me3 and H3K9ac, are defined as ELE-TSSs. CAGE signals located at the 5′-end
of the genes or transcripts with a relatively weak signal, but supported by epige-
netic features including H3K9ac, H3K4me3, H3K36me3, RNA-sequencing and open-
chromatin accessibility, are classified as low-confidence (LC) gene TSSs. **g** Donut
plot showing the distribution of CAGE clusters in genic and intergenic regions in
four tissues. **h** Scatter plot showing the transcription levels in intergenic regions as
determined by CAGE-seq (x-axis) and total RNA-seq (y-axis) data. Each dot repre-
sents a CAGE cluster. The lines represent contours. **i** Heatmaps of the CAGE-TSSs
surrounding the ELEs defined by the H3K4me3 and H3K9ac peaks. The heatmaps
present the signal densities for CAGE-TSSs (left) and H3K4me3 or H3K9ac (right);
peaks are ordered according to the increasing length of ELEs. Source data are
provided as a Source Data file.

observed that TE-embedded ELE-RNA TSSs were hypo-methylated
(Supplementary Fig. 15) and harbored active epigenetic environments
comparable to ELE-RNAs in non-TE regions (e.g., enrichment of
H3K4me3, H3K9ac, open chromatin, and CpG islands) (Fig. 2g–i and
Supplementary Fig. 16), indicating that these TE-embedded ELEs are
potentially active. Comparison of the three subgenomes revealed that
subgenome A contributed more abundant TE-derived ELEs and nas-
cent transcripts (Supplementary Fig. 17).

### Specific expansion of RLG_famc7.3 in subgenome A created hundreds of spike-specific ELE-RNAs

ELE-RNAs are mainly produced by Gypsy-type LTR TEs, without sig-
nificant differences with genomic TE distribution (Fig. 3a). TEs were
classified into different families based on sequence similarity (see
Methods)[30,31]. Among the TE families highly abundant in common
wheat genome, the top enriched TE subfamily contributed to ELE-RNA
is RLG_famc7.3, which produces 22% of TE-initiated ELE-RNAs (Fig. 3a,
b). This subfamily was mainly detected in subgenome A (Supplemen-
tary Data 5), whose expansion occurred in the diploid progenitor after
the divergence from the A-B-D common ancestor (Fig. 3c and Sup-
plementary Fig. 18). The transcription of RLG_famc7.3-derived ELE-
RNA exhibited the most significant tissue specificity (i.e., pre-
dominantly in spikes) among the top abundant TE families contribut-
ing to ELE-RNAs (Fig. 3d, e). To elucidate the regulation of ELE-RNA
transcription, we searched for the over-represented TF-binding motifs
surrounding TSSs of RLG_famc7.3 TE-embedded ELE-RNAs. The most
enriched TFBSs included BPCs and RAMOSA1, which are AG-rich
motifs (Fig. 3f) affecting inflorescence branching and floral
development[32–34]. These motifs were more abundant in RLG_famc7.3-
initiated ELE-RNAs than in the RLG_famc7.3 TEs (Fig. 3g–h). Thus, the
expansion of RLG_famc7.3 and the acquisition of specific TF-binding
motifs likely resulted in spike-specific ELE-RNAs.

### RLG_famc7.3-embedded ELEs regulate spike specificity

In order to determine whether the spike-expression of RLG_famc7.3-
embedded ELEs have functional influence on wheat development, we
compared the spike-specific ELE-RNAs initiated by RLG_famc7.3 with
loci controlling agronomic traits detected by genome-wide association
study[35]. The RLG_famc7.3-initiated ELE-RNAs were mainly enriched at
loci regulating spike development and rust resistance (Fig. 4a and
Supplementary Data 6). We knocked down one RLG_famc7.3-derived
ELE-RNA higher expressed in spike using an RNA interference strategy
in the cultivar 'JW1' (Supplementary Data 7), which is easier for trans-
genic manipulation. Compared with the negative control transgenic
lines (CK), no significant difference was observed in vegetative growth.
Apparent enrichment of small RNAs and reduced ELE-RNA expression
was detected in the target loci in T1 generation (Supplementary
Figs. 19–20). The spike statistics are performed in the T1 generation.

The spikes of knockdown lines developed abnormally, with a slightly
longer rachis (Fig. 4b, c) and a relatively greater distance between
spikelets (Fig. 4d). The dispersed spikes of the knockdown lines did not
affect the total number of grains (Fig. 4e), but they were associated
with a slight decrease in the kernel weight (Fig. 4f), suggesting that
RLG_famc7.3 likely influences the wheat yield. The spike phenotype
was consistent between the $T_0$ and $T_1$ knockdown lines (Supplemen-
tary Fig. 21). An examination of the expression of the genes potentially
targeted by the RLG_famc7.3-embedded ELEs revealed that 13% of
these genes were expressed at lower levels in the knockdown lines than
in the CK plants (Fig. 4g and Supplementary Data 7). The data were
correlated among independent knockdown lines with longer
rachis (Supplementary Fig. 22). Taken together, RLG_famc7.3-initiated
ELE-RNAs regulate target gene expression and common wheat devel-
opment in a spike-specific manner.

## Discussion

TEs are an abundant resource for the creation of new regulatory REs in
both mammals[36–41] and plants[42–46], especially in common wheat with
extremely large genome harboring abundant TEs[7]. After differentia-
tion from a common ancestor, three diploid progenitors undergo
active TE birth and death[5,31], created abundant and diversified reg-
ulatory elements, which potentially endowed common wheat with
versatile strategies for coping with internal or external changes[47,48].
However, majority of TE-affected expression divergence may be tran-
scriptional drift without phenotypic consequences and a causal link is
missing. Here, we predicted the function of TE-derived ELEs via pro-
filing dynamic transcriptome of ELE-RNAs from different develop-
mental stages, detected a cohort of TE-initiated ELE-RNAs that
specifically expand in subgenome A, and demonstrated the direct
impact on regulating spike specificity. Despite that knocking down of
one ELE-RNA resulted in weak changes in spike development, which is
a complex trait cooperatively regulated by many loci, the strategy and
findings help to elucidate the causal effects of TEs on agronomic traits,
providing insight into the direct regulatory function of numerous TEs
in common wheat, and their contribution to developmental specificity
and polyploid plasticity (Fig. 5).

It is noteworthy that the spike phenotypes of diploid progenitors
are highly diverse[49]. Both phenotypic and evolutionary studies inves-
tigating subgenome bias indicated that the common wheat spike
phenotype is mainly contributed by *Triticum Urartu* (Tu), the diploid
progenitor of subgenome A[50,51]. In the present study, the knockdown of
RLG_famc7.3-initiated ELE-RNAs specifically expanded in Tu resulted in
aberrant expression of spike-specific genes and spike development.
Despite the phenotypic changes are relatively weak, these findings
suggest that the distinct spike phenotypes across subgenomes are
partially due to subgenome-specific TE-initiated ELEs, providing clues
for elucidating the impacts of the the long-overlooked highly

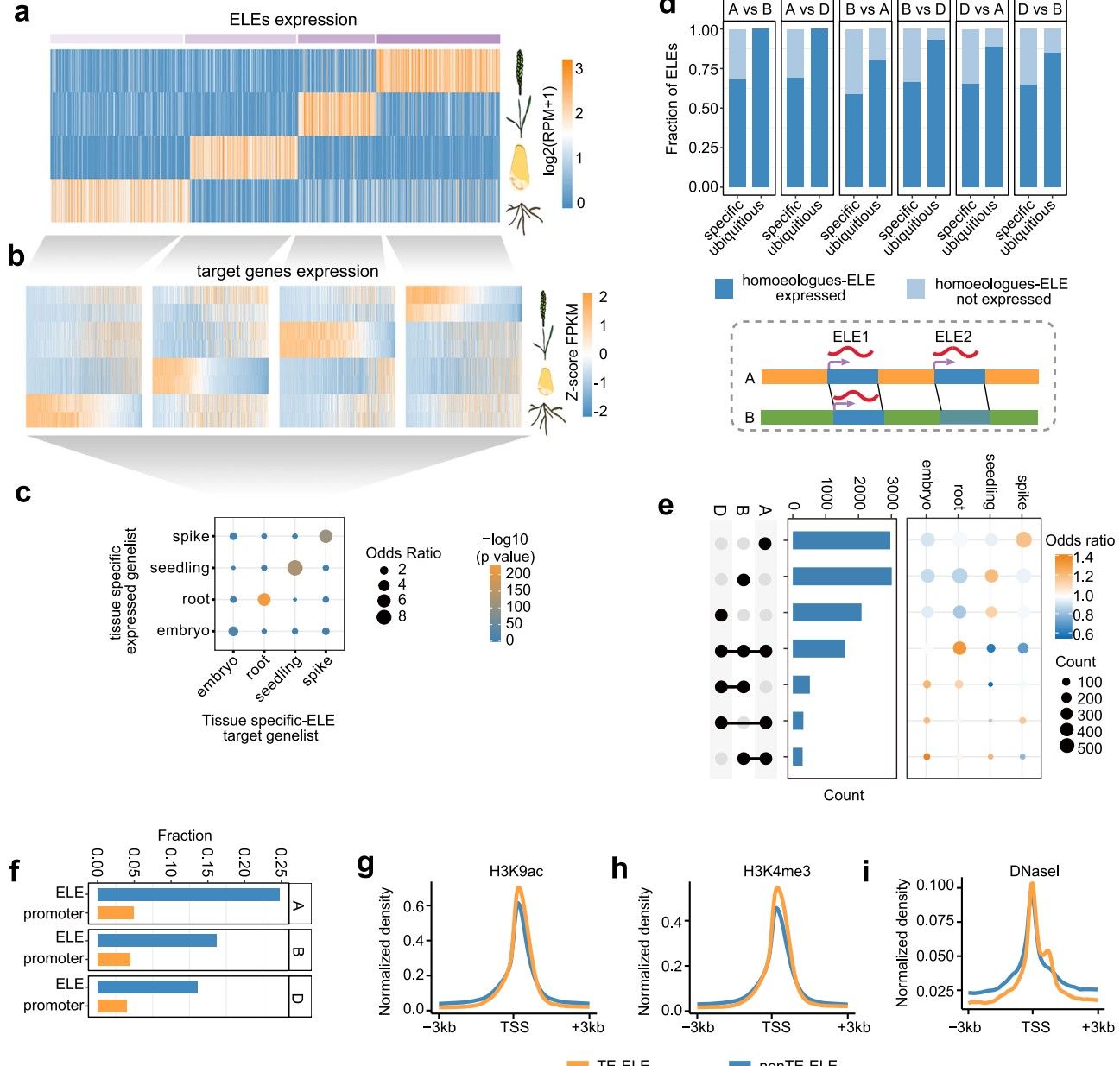

**Fig. 2 | Tissue-specificity of subgenome-biased enhancer-like-element transcripts. a** Heatmap showing tissue-specific ELEs expression. RPM, reads per million mapped reads. **b** Heatmap showing the z-scored expression of the predicted target genes for the corresponding ELEs. **c** Enrichment of the tissue-biased genes and tissue-specific ELE targets. The p-value and odds ratio were determined by two-tailed Fisher's exact test with the total genes as the background. **d** The bar plot illustrates the expression of homoeologous regions corresponding to tissue-specific and ubiquitously expressed ELEs. Pairwise comparisons were made for the A, B and D subgenomes. Diagrams within dashed boxes illustrate categories of ELEs based on homology and expression level. Homoeologous ELEs within collinear regions between subgenomes A and B are connected by black lines. In this example, the homoelog of ELE1 is expressed but the homoelog of ELE2 is not expressed. **e** Left panel, the abundance of subgenome-common and -specific ELE-RNAs. Right panel, enrichment of each ELE-RNA group in tissue-specifically expressed ELE-RNAs. The total ELE-RNAs were used as background. **f** Fraction of CAGE-detected gene and ELE-RNA TSSs initiated from TEs in the A, B, and D subgenomes. **g–i** Epigenetic profiles 3 kb up- and down-stream of TE- and non-TE-initiated ELE-TSSs, including the active histone marks H3K9ac and H3K4me3, and open chromatin characterized by DNaseI-seq. Source data are provided as a Source Data file.

abundant TEs in wheat. Compared to other TE-derived ELEs, RLG_famc7.3-derived ELEs are less conserved between diploid progenitor and hexaploid common wheat and are largely conserved across hexaploid and tetraploid wheat (Supplementary Fig. 23), indicating this subfamily of TEs possibly underwent some sort of selection during domestication following hexaploidization, and were fixed in polyploid wheat.

Whether TE-embedded TFBSs are mainly responsible for driving TE propagation or are co-opted for the regulation of host genes is

unclear[37]. Recent research suggested that germline activity increases the likelihood of TE inheritance and expansion[52]. We performed an ALE-seq (short for amplification LTRs of extrachromosomal linear DNA (eclDNA) followed by sequencing)[53] to detect the transpositional activity of TEs in nine common wheat tissues, as well as in a common wheat population and in the diploid progenitor of subgenome A (Supplementary Fig. 24). Almost no eclDNA was detected for RLG_famc7.3 (Supplementary Data 8). We analyzed the results of the previously published dual luciferase reporter assay[17] and found that

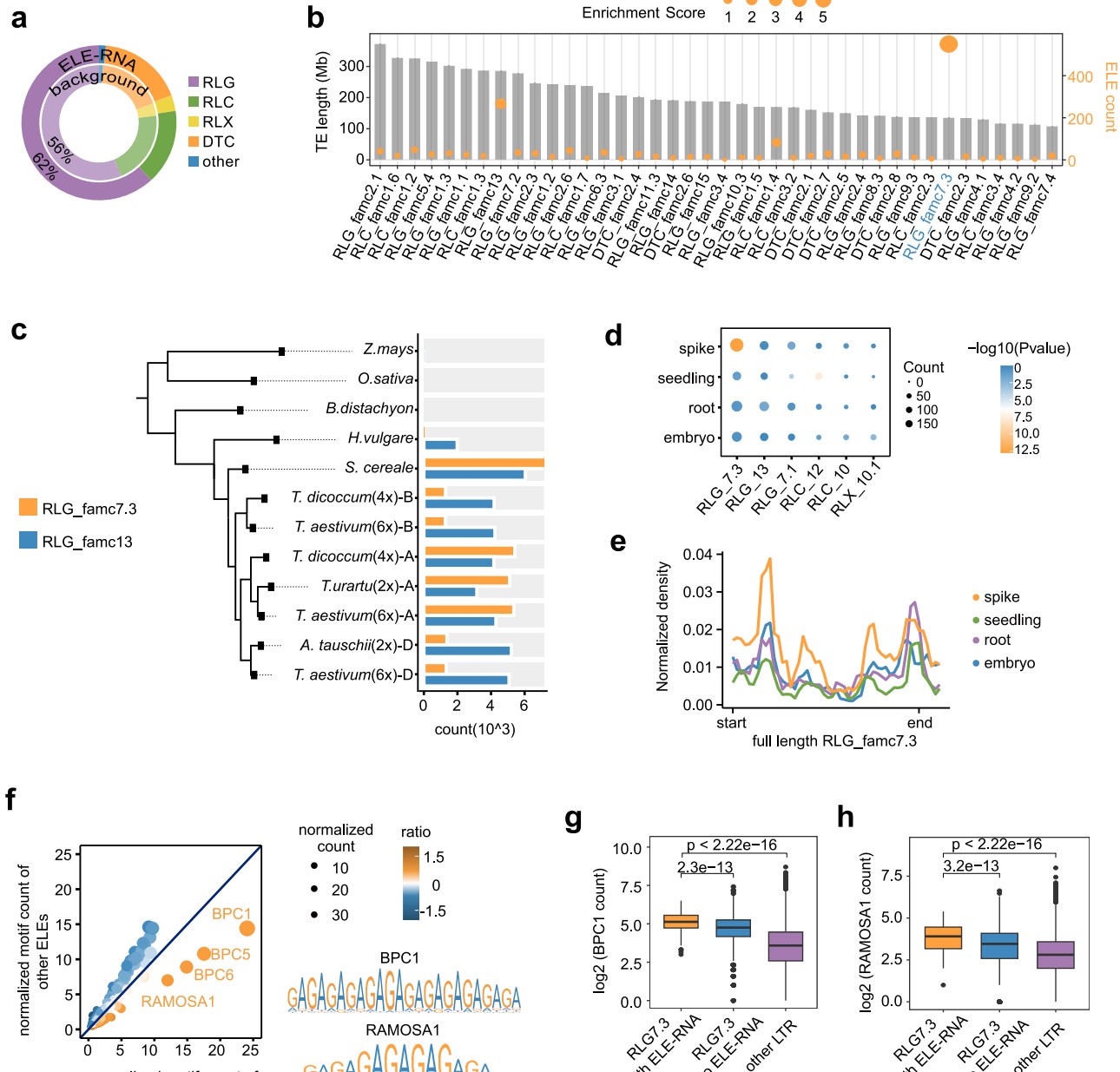

**Fig. 3 | Subgenome A-specific RLG_famc7.3 expansion and TFBS acquisition created spike-specific ELE-RNAs. a** Donut plot showing the distribution of TE super families in the TE-ELEs and the overall genome. **b** The plot displays the top 40 most abundant TE subfamilies. The height of each bar represents the TE length of the corresponding subfamily. The height of each point represents the number of ELEs embedded in each subfamily. The size of each point represents the enrichment score of the corresponding TE subfamily generating ELE-RNAs. The whole genome was used as background. **c** Abundance of RLG_famc7.3 and RLG_famc13 in wheat species of different ploidy levels and relevent grasses. Phylogenetic analysis suggests that RLG_famc7.3 has expanded in *Triticum Urartu* (AA). **d** Enrichment of tissue-biased transcription of ELE-RNAs initiated by different TE families. Two-tailed Fisher's exact test was used to test the enrichment of TE-ELE-RNAs and tissue specific ELE-RNAs. The overall ELE-RNAs as background. The TE subfamilies

contributing the most abundant ELEs are presented. **e** Average expression profiles of the full-length RLG_famc7.3 members in different tissues as reflected by the total RNA-seq density, which was normalized by the LTR length. **f** Scatter plot comparing the motif abundance in RLG_famc7.3-initiated ELEs and in other ELEs. The BPCs and RAMOSA1 are preferentially present in RLG_famc7.3-initiated ELEs. **g–h** Box plots showing the distributions of BPC1 (**g**) and RAMOSA1 (**h**) motif abundances for RLG_famc7.3 with ($n = 167$) or without ($n = 7841$) ELE-RNAs. Other LTR sequences were selected as the controls ($n = 450174$). Full-length LTR retrotransposons were used for the motif scan. The significance of the differences was determined according to the two-tailed Welch two-sample t-test. Horizontal lines in boxplots show median, hinges show IQR, whiskers show 1.5 × IQR, points beyond 1.5 × IQR past hinge are shown. Source data are provided as a Source Data file.

out of the 22 active fragments, 7 of them were derived specifically from RLG_famc7.3 and exhibited enhancer activity. Thus, it is likely that a subset of TEs generated active enhancers and mediated reciprocal adaptation between TEs and hosts.

Neither the origin nor the function of eRNAs in plants is clear. The current study provided multiple lines of evidence supporting the

functional relevance of TE-initiated ELE-RNAs with host gene regulation and developmental specificity. First, we determined that capped ELE-RNA transcription specificity is predictive of the impact of ELE specificity in transcriptional regulation (Fig. 2a–c). Second, we revealed the active epigenetic architecture and the decrease in the repressive DNA methylation surrounding ELE-RNAs (Fig. 2g–i and

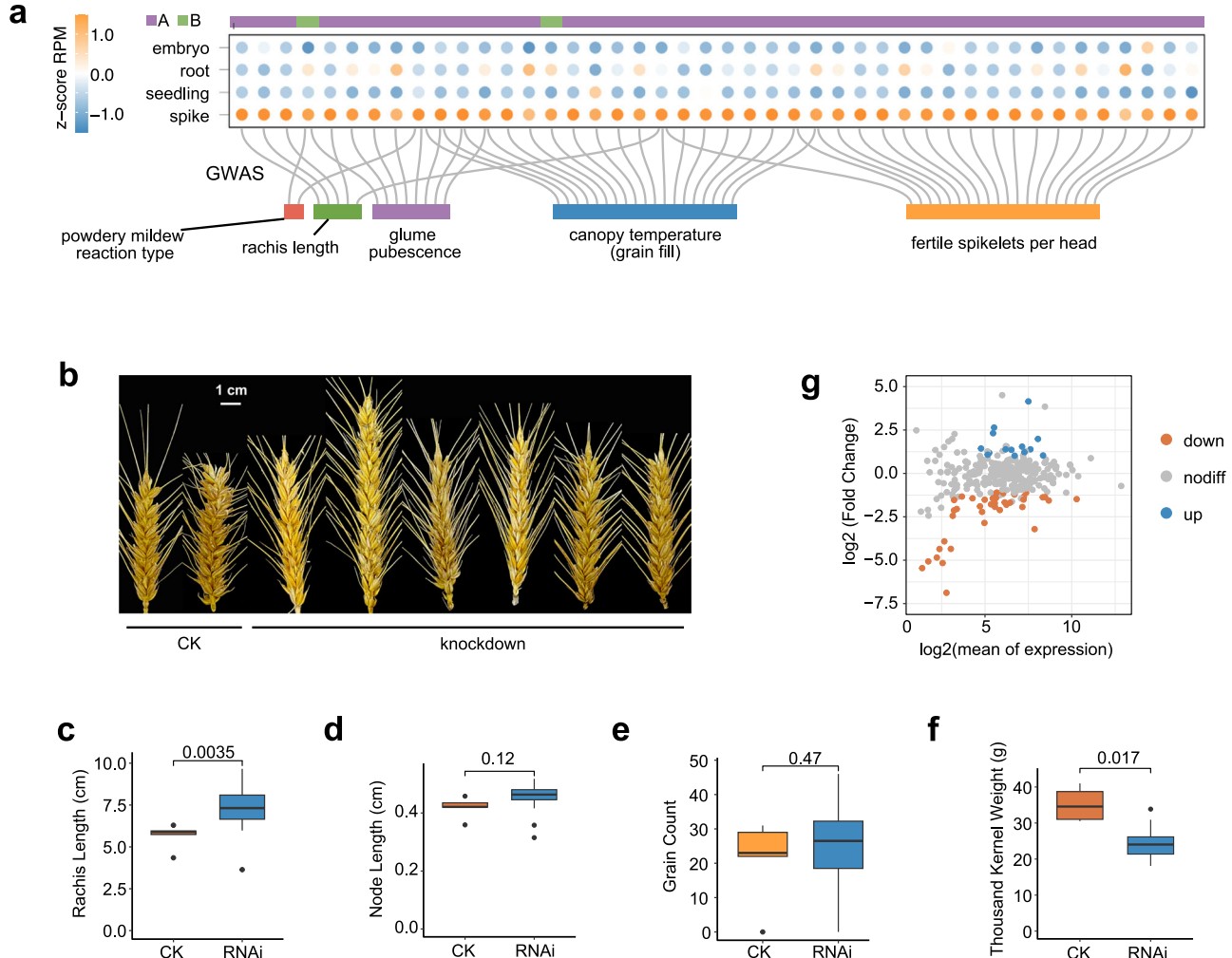

**Fig. 4 | Functional relevance of RLG_famc7.3-initiated spike-specific ELE and development. a** GWAS traits with enriched causal variants surrounding spike-specific ELE-RNAs initiated by RLG_famc7.3. Spike-related terms are provided. All enriched terms are listed in Supplementary Data 6. **b** Mature spikes of the $T_1$ generation knockdown lines and CKs. The main spikes of each plant are presented. The phenotypes of the $T_0$ generation are presented in Supplementary Fig. 21. **c–f** Comparison of the spike phenotypes between the knockdown lines and CKs. All of the spikes from each plant were examined ($n = 5$ for CK; $n = 22$ for RNAi). The

significance of the differences was determined according to the two-tailed Welch two-sample t-test. Horizontal lines in boxplots show median, hinges show IQR, whiskers show $1.5 \times$ IQR, points beyond $1.5 \times$ IQR past hinge are shown. **g** MA plot of the differences between the knockdown lines and CKs in terms of the expression of the genes targeted by RLG_famc7.3-initiated ELEs. Differentially expressed genes with $|\log_2(\text{fold-change})| > 1$ and $P < 0.05$ are in orange (downregulated genes) and blue (upregulated genes). The other genes are in gray. Source data are provided as a Source Data file.

Supplementary Fig. 15). Third, we detected the considerable enrichment of flowering- and floral development-related TF-binding motifs in spike-specific ELE-RNAs (Fig. 3f–h). Fourth, we detected the changes in spike development (Fig. 4 and Supplementary Fig. 21) following the knockdown of RLG_famc7.3-initiated ELE-RNAs. Similar phenotype was observed when the targeted RLG_famc7.3-initiated ELE-RNA was over-expressed (Supplementary Fig. 25), possibly due to co-suppression with increased small RNA (Supplementary Fig. 25). However, we still can't distinguish whether the impacts on spike development in RNA interference lines are due to ELE-RNAs or directly due to ELEs, since in addition to 21 nt small RNAs commonly repressing target RNA expression, we also detected abundant 24 nt small RNAs which may contribute to RNA-dependent DNA methylation. Consistently, apparent DNA methylation was detected in the RNA interference targeted regions by Chop-PCR based on partial digestion by methylation-sensitive restriction enzymes followed by PCR amplification[54] (Supplementary Fig. 26). Together, the early production of ELE-RNAs and close association with ELE activity makes them an excellent marker for predicting the functions of ELEs without identifying their exact target genes. Further genetic

engineering targeting the functional regulatory elements or their transcripts provides greater flexibility for the spatiotemporal optimization of agronomic traits.

## Methods

### Plant materials and growth conditions

Common wheat (*Triticum aestivum* cultivar 'Chinese Spring') seeds were surface-sterilized via a 10-min incubation in 30% $H_2O_2$ and then thoroughly washed five times with distilled water. The seeds were germinated in water for 3 days at 22 °C. The germinated seeds with residual endosperm were transferred to soil (1:1:3 mixture of vermiculite: perlite: peat soil) or the Hoagland solution and grown under 16 h light/ 8 h dark conditions at 22 °C in the greenhouse. The seedlings (above-ground parts) in the soil were harvested after 9-day growth. The roots in the Hoagland solution were harvested after 9-day growth. The spikes at the booting stage (Feeke 10) were harvested. The fresh immature embryos (14 days post anthesis) were isolated and either frozen in liquid nitrogen for RNA isolation or Bisulfite-seq and directly vacuum-infiltrated with a formaldehyde cross-linking solution for ChIP-seq assay. Wheat cultivar 'JW1' was used to do the genetic

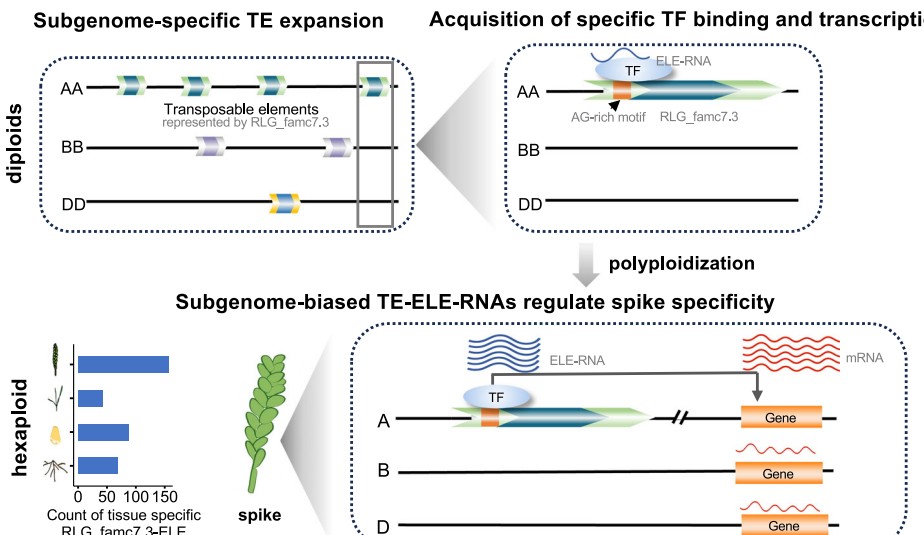

**Fig. 5 | Model illustrating the evolution of TE-initiated ELEs generate the subgenome-biased spike specificity in common wheat.** Lineage-specific expansion of TE subfamilies contributed to subgenome-divergent ELEs. Further acquisition of spike-specific TF-binding sites resulted in the spike-specific ELE-RNA transcription, which is associated with regulating subgenome-biased spike specificity in common wheat.

transformation and grown under 16 h light/8 h dark conditions at 22 °C in the greenhouse.

### CAGE-seq library construction and sequencing

Total RNA was treated with RQ1 DNase (promega) to remove DNA. The quality and quantity of the purified RNA were determined by measuring the absorbance at 260 nm/280 nm (A260/A280) using Nanodrop One (Thermo). RNA integrity was further verified by 1.5% agarose gel electrophoresis. For each sample, 1 μg of total RNA was used for CAGE-seq library preparation. The total RNA was treated with T4 polynucleotide kinase (NEB) at 37°C for 30 min and subsequently digested with Terminator 5´-Phosphate-Dependent Exonuclease (Ambion) at 30°C for 30 min to enrich the capped mRNA. Next, reverse transcription was performed with RT primer harboring a 3'-adaptor sequence and randomized hexamer. Subsequently, the 5'adaptor harboring three additional rG at the 3'terminus was added to the RT reaction and incubated for another 30 min to allow template switching and tagging. The cDNAs were treated with Exonuclease I (epicentre) to digest the primers and purified. Then the cDNAs were amplified with PCR primers (Illumina) and the PCR products corresponding to 250–500 bps were purified and quantified, then stored at −80 °C until sequencing. For high-throughput sequencing, the libraries were applied to Illumina Novaseq 6000 system for 150 bp paired-end sequencing.

### RNA-seq, ChIP-seq and Bisulfite-seq library construction and sequencing

For RNA-seq, more than 2 μg total RNA was used to prepare each sequencing sample. Total RNA-seq libraries were constructed and sequenced by Novogene (Beijing, China). ChIP-seq assays were performed using specific antibodies to H3 trimethyl-Lys 27 (Millipore, 07360, Upstate, USA), H3 trimethyl-Lys 4 (Abcam, ab8580, Cambridge, England) and H3 acetyl-Lys 9 (Millipore, 07352, Upstate, USA). The dilution is 1:100. For each ChIP-seq assay, more than 10 ng ChIP DNA was used to prepare each sequencing sample[55]. The 2.2 μg of DNA extracted from each sample was used to prepare the bisulfite sequencing samples. Libraries were constructed and sequenced by Genenergy Biotechnology Co. Ltd. (Shanghai, China) and Novogene (Beijing, China). The libraries were sequenced with the HiSeq X Ten system (Illumina, San Diego, California, USA) to produce 150-bp paired-end reads. Overall, we generated RNA-seq, ChIP-seq and Bisulfite-seq data for the root, spike, and embryo in this study. The seedling data used in the analysis was obtained from a previously published study[26].

### Processing of CAGE-seq data

For CAGE-seq data, only the R1 reads were kept for mapping. Cutadapt (version 1.18)[56] was applied to trim 3' adaptors and 9 bases at the 5'end which included randomized hexamer and three additional rG. Bases with low-quality scores (<20) and short reads (length <20) were eliminated. We then used SortMeRNA (version 2.1b) program[57] to remove the reads originating from chloroplast, mitochondria and rRNA. The remaining clean reads were mapped to the International Wheat Genome Sequencing Consortium (IWGSC) reference sequence (version 1.0) with Bowtie2 (version 2.3.5)[58]. Only unique mapped reads were used for analysis. To reduce false positive 5' end reads due to library construction, we aligned total reads without removing the 9 bases at the 5' end and the reads aligned with less than 3 mismatches were eliminated. The 5' coordinates of R1 reads were used as the position of the transcription start sites. R package CAGEr (version 1.28.0)[59] was performed to call the CAGE-tag clusters (CTC). Each tissue was called separately. For the tissue-specific clusters, only CTCs supported by at least two samples were kept. TSSs of four tissues were then merged into a larger and unique TSS database. The TSSs with lengths longer than 10 bp were defined as broad TSSs.

### CAGE-TSS annotation

To classify the CTCs and assign the protein-coding TSS to the corresponding genes, we first used the RNA-seq of 10 wheat tissues to assemble transcripts and combined the IWGSC ref1.1 to generate a comprehensive coding gene model database. The CTC located in the 5'UTR region or 500 bp upstream of the TSS was assigned to that gene and defined as gene-TSS (Supplementary Fig. 1d). CTCs more than 3 kb away from the gene models were defined as intergenic TSS. Intergenic TSSs with active enhancer markers for H3K4me3 or H3K9ac were defined as ELE-TSSs (Supplementary Fig. 1d).

In addition to the CTCs detected above, we also noticed weak CAGE signals around or within genes, some of which may be authentic start sites for transcription. We integrated epigenetic data, nascent RNA-seq and CAGE-seq data to characterize chromatin states based on Hidden Markov Model using chrHMM (version 1.23) (Supplementary Fig. 1a, b)[60]. Genome bin size was set to 500 bp to quantify

modifications and transcript levels. The "LearnModel" algorithm was used to classify the genome into 15 chromatin states based on signal combinations. Data from seedlings were used for training. This dataset contains epigenetic profiles, including 8 chromatin modification ChIP-seq data, Pol-II ChIP-seq, and DNaseI-seq, and transcriptional data, including GRO-seq, pNET-seq, CAGE-seq, and RNA-seq.

Different states represent distinctive chromatin architectures and are reflected by the combinatorial patterns of multiple marks. State 5 is characterized by the enrichment of CAGE signals, nascent transcriptome signals (enriched at TSS), and promoter-associated epigenetic modifications H3K4me3 and H3K9ac. States 2–3 and states 6–7 are enriched for RNA-seq and transcription elongation-related epigenetic markers H3K36me3 and H3K4me1 (Supplementary Fig. 1a). The transition probabilities from state 5 to state 2–3 and state 6–7 are higher (Supplementary Fig. 1b). These results indicate that state 5 contains a subset of active TSSs. 44,930 weak CAGE tag clusters located in state 5 were recognized as low-confidence gene TSS (Supplementary Fig. 1c).

### Processing of ChIP-seq, Bisulfite-seq and RNA-seq data
The sequencing reads were trimmed with the Trim Galore (version 0.6.4) and cutadapt[56]. For RNA-seq, clean reads were aligned with HISAT2 program (version 2.1.0)[61]. SortMeRNA (version 2.1b)[57] was used to remove reads originating from chloroplast, mitochondria and rRNA. The featureCount program of the Subread package (version 1.5.3)[62] was used to determine the RNA-seq read density of the high-confidence genes in the IWGSC RefSeq genome assembly (version 1.1)[5]. The DEseq2 R package[63] was used for detecting differentially expressed genes based on the following criteria: |log2 fold-change|> 1 and $P < 0.05$. The StringTie program (2.1.4)[64] was applied to assemble the potential transcripts using 10 RNA-seq samples including different developmental states. Transdecoder (version 5.5.0) (https://github.com/TransDecoder/TransDecoder/wiki) and hmmsearch (version 3.2.1)[65] were applied to annotate the transcripts. For ChIP-seq data, clean reads were aligned with bwa program (version 0.7.17-r1188)[66]. The MACS2 (version 2.1.1) program[67] with parameters "--nolambda --nomodel" was used to identify the peaks. For Bisulfite-seq data, the clean reads were aligned by the Bismark program (version 0.19.0)[68] with the default setting, and non-unique alignments were removed. The extent of the cytosine methylation was determined by the tool bismark_methylation_extractor implemented in the Bismark program. Next, the methylation ratio of cytosines was calculated as the number of mCs divided by the number of reads covering the position. The negative values in the data indicate negative strands. Bases covered by less than three reads were considered low confidence positions and their methylation ratios were not recorded.

### 5'RACE validation
CAGE-TSSs inconsistent with the IWGSC annotation were randomly selected for 5'RACE validation. Total RNA was extracted from the leaf for reverse transcription. PolyC was added to the 3'end of cDNA. The adaptor-poly (G) and adaptor-nest-PCR primers were used for nested PCR. The PCR products were sent for Sanger sequencing. All primers used in this study are listed in Supplementary Table 1.

### Motif analysis
For ELEs motif enrichment analysis, regions 1 kb upstream and downstream of the TSSs were selected for motif scan. The motifs from JASPAR-plant were scanned by the Find Individual Motif Occurrences program of the MEME software toolkit[69] with the parameter "--max-stored-scores 1000000". The position and the number of each motif in each TSS were then recorded and summed. The background regions were randomly generated from the genome with the same length distribution. For motif comparison between ELEs and promoters or between RLG_7.3-ELEs and other TE ELEs, the number of motif

occurrences was normalized by the total number of TSSs of the given group. For motif comparison between different types of RLG_7.3 TEs, the annotated full-length RLG_7.3 TEs were used to scan the motif. The parameters were the same as above.

### Kmer-SVM training for TSS classification
The R package "gkmSVM"[70] was applied for training an SVM model to predict the type of TSSs. Sequences 250 bp up- and down-stream of TSS were used. 80% of ELE-TSSs and the same count of gene-TSSs were taken as the training dataset and the remaining TSS were used as the prediction dataset. The output of the training and prediction datasets were combined to find the cutoff for the prediction model. There were 1588 ELE-TSSs could not be predicted correctly. Blastn was used to align these ELE-TSSs with all gene-TSS sequences to obtain the alignment ratio.

### Connect target genes to ELEs
The ELE-TSSs and gene-TSSs within 2 M were chosen as pairs. All CAGE-seq replications of 4 tissues were used to calculate the CAGE signal (represented by RPM) correlation. The RP score of a gene was defined as the sum of the distance-weighted ELE contributions and calculated using the following formula[71]:

$$RP = \sum_{i=1}^{k} 2^{\frac{-di}{d0}} \qquad (1)$$

The parameter $d_O$ is the decay distance of the weight function and was set to 100 kb here. The $k$ ELEs near the gene (within the distance of $20 * d_O$) were used in the calculation, and $d_i$ is the distance between the $i$th ELE and the gene. Genes within 2 M from the ELEs with CAGE signal correlation greater than 0.5 and the RP score in the top 50% were determined as predicted targets.

### Hi-C data analysis for promoter and ELEs interactions
Hi-C data was downloaded from the NCBI BioProject database (accession number GSE133885[9]. The data mapping and processing were performed as reported[8]. The contact matrix in 50 kb bin was generated with Juicer[72]. The interaction strength between ELEs and target genes was considered as the interaction of the corresponding bins.

### ELEs comparison across subgenomes
To compare ELEs across subgenomes, we first obtained the homoeologous sequence of ELEs in other subgenomes. Due to the presence of many repetitive sequences in wheat genome, the syntenic blocks of the ELEs between subgenomes were detected to narrow down the search space. Specifically, MCScanX package[73] was applied to obtain the collinear genes between subgenomes. The regions between collinear genes were treated as syntenic blocks. ELEs in regions with syntenic block in other subgenomes were extracted for statistics. Blastn was used to align the ELE-TSS and its downstream 200 bp sequence to the corresponding syntenic regions and the evalue cutoff was set to 0.01. Aligned ELEs were categorized as "with homology sequence", otherwise as "without homology sequence". Next, ELEs were classified as with or without homology ELEs based on whether the homologous fragments have active chromatin modification peaks.

### TE annotation
TEs were annotated by CLARITE[5]. Specifically, the whole genome of 'Chinese Spring' was searched based on sequence similarity using a high-quality TE databank called ClariTeRep. This approach allows the identification of the overall positions, types and families of TEs. ClariTeRep is a database containing sequences present in TREP, a curated library of Triticeae TEs from all three subgenomes, as well as TEs manually annotated in a previous pilot study[30] on chromosome 3B.

This resulted in the identification of 14 superfamilies, 451 families and 516 subfamilies. For example, the superfamilies include Gypsy (RLG), Copia (RLC), and CACTA (DTC), the families include RLG_famc7 and DTC_famc5, and RLG_famc7.3 and RLG_famc7.1 are subfamilies of RLG_famc7. The full-length LTRs were annotated by the LTRharvest suite (version 1.5.9)[74]. The elements in TEs were distinguished by LTRdigest implemented in LTRharvest[75]. The TE families and subfamilies were designated according to the rules of the ClariTeRep database[30,31]. The ELE-TSSs embedded in the annotated TEs were defined as TE-ELEs.

## TE insertion time calculation
The TE insertion time was determined on the basis of the divergence between the 5′ and 3′ LTRs[5]. Briefly, the sequences of long terminal repeat regions in both ends of full-length LTR were extracted and aligned by prank (v.170427)[76]. The distance was calculated by emboss dismat (version 6.6.0.0)[77] and corrected by applying the Kimura 2-parameter. The insertion time was estimated using the formula:

$$time = distance/(mutation\ rate*2*100) \qquad (2)$$

with a mutation rate of $1.3*10^{-8}$ per base pair per generation.

## ALE-seq library construction, sequencing and processing
ALE-seq is a technology designed to detect active LTR retrotransposon by amplifying eclDNA. The amplification involves two reactions: in vitro transcription and reverse transcription(Supplementary Fig. 24a)[53]. Plant materials of seedling, root, embryo and spike same as above and other samples (Supplementary Data 8) were used to extract DNA and construct ALE-seq libraries[53]. 500 ng genomic DNA was used for adapter ligation and the adapter-ligated DNA was purified by VAHTS DNA Clean Beads (Vazyme, N411). In vitro transcription reactions were performed using an Ambion MEGAscript T7 RNA synthesis kit, the RNA product was purified and quantified. Then 3ug purified RNA was subjected to reverse transcription with pooled PBS_Illumina primers. After the reverse transcription reaction, 1 μl of RNase A/T1 (Thermo Fisher) was added to digest non-templated RNA. Finally, Single-stranded first strand cDNA was PCR-amplified by 25 cycles according to used adapter primers and the PCR product was purified by VAHTS DNA Clean Beads. Libraries were sequenced by Novogene (Beijing, China) to generate 150 bp paired-end reads. The custom reverse transcription primers are shown in Supplementary Table 2. Cutadapt[56] was used to trim the raw sequencing reads. Clean reads were aligned to the IWGSC reference sequence (version 1.0)[5] with Bowtie2 (version 2.3.5)[58]. IGV[78] was used to visualize the sequencing data. For the alignment-based approach, the forward and reverse reads were merged to yield the full-length fragment sequences, then calculate the coverage in the long terminal regions of LTR retrotransposon (Supplementary Fig. 24b). Only the full-length fragments started from the 5′ of the long terminal regions were counted (Supplementary Fig. 24c).

## GWAS data analysis
The GWAS results were downloaded from T3 database (https://wheat.triticeaetoolbox.org/genome/gwas.pl)[35]. The variant loci were extended based on previously published linkage disequilibrium decay distance[79]. Fisher's exact test was used to determine the enrichment of the causal variants of GWAS traits in spike-specific expression ELEs.

## Plant transformation for RNAi silencing and phenotyping
The cultivar 'JW1' was used for RNA interference. Before experimental validation, we systematically compared the sequences and epigenetic profiles of ELEs between 'JW1'and CS, to ensure the presence and activity of targeted ELEs. For the ELEs with conserved sequences between the two cultivars, the epigenetic activity is highly consistent;

352 of 411 spike active RLG_famc7.3 ELEs also had active chromatin states in JW1 spikes (Supplementary Fig. 27a).

The sequences downstream 1–200 bp and 200–400 bp of the RLG_famc7.3 ELE- TSS located in JW1 and have identical sequences in CS (Supplementary Fig. 27b) were synthesized and cloned into pEXT06/g to construct RNAi silencing vector. The plasmid was then transformed into wheat cultivar 'JW1' via Agrobacterium-mediated transformation by Wimi Biotechnology (http://www.wimibio.com/). The primer sequences used in this study are shown in Supplementary Table 3.

A total of 38 transgenic lines and WT were grown in the greenhouse. The spike at the booting stage (Feeke 10) of 3 WT and 4 transgenic lines of the T0 generation were taken to extract RNA. Libraries were constructed and sequenced by Hanyubio Co. Ltd. (Shanghai, China) to generate 150-bp paired-end reads. The RNA-seq data were processed with the same method as above. The differentially expressed genes were defined based on the following criteria: |log2 fold-change |> 1 and P < 0.05.

A total of six transgenic T1 and WT lines were grown in the greenhouse at 22 °C under 16 h light/8 h dark conditions for phenotyping. Each line was planted with 1–3 seeds and the total mature spikes were harvested to characterize the spike and yield features (Supplementary Table 4). ImageJ was used to measure the scaled pictures. The rachis length does not include awns. The node length is the rachis length divided by the number of spikes.

## smRNA-seq, DNA methylation measurements and RT-qPCR at RLG_famc7.3-ELE loci to evaluate the efficiency of RNAi silencing
For smRNA-seq, the spike at the booting stage (Feeke 10) of 2 transgenic lines of the T1 generation and WT were taken to extract 4 μg RNA. Libraries were constructed and sequenced by Novogene (Beijing, China) to generate 50-bp single-end reads. The reads were trimmed with cutadapt(version 1.18)[56] to remove Illumina adapters and low-quality bases (quality score < 30). Shortstack (version 3.8.5)[80] was utilized to map the 21-nt and 24-nt clean reads. Initially, default parameters were applied to identify smRNA clusters generated by the RNAi strategy. The reads mapping to the active RLG_famc7.3 were subsequently isolated and remapped using bowtie (version 1.1.2)[81] with the parameter "-a -l 5 -e 100" to allow for mismatches. This approach facilitated the identification of additional RLG_famc7.3-ELEs that could be targeted by enhanced smRNA. Subsequently, Chop-PCR (methylation-sensitive enzyme digestion followed by PCR) was designed for the primary targeted RLG_famc7.3-ELE and one of the other RLG_famc7.3-ELEs that could be targeted by the enhanced smRNA. Chop-PCR is a targeted DNA methylation detection technique that uses partial digestion by methylation-sensitive restriction enzymes (MSREs) followed by PCR amplification. The presence of cytosine methylation at the cleavage sites of the MSREs protects the DNA against digestion and therefore can be amplified using PCR[54].

For Chop-PCR, the genomic DNA of the same material was extracted using CTAB method to evaluate the DNA methylation level of the RLG_famc7.3-ELEs. 100 ng of genomic DNA was incubated 30 mins with the methylation-sensitive restriction enzyme NlaIII (NEB). The digested DNA was used to amplify the smRNA targets by semi-quantitative RT-PCR. Non-digested genomic DNA was simultaneously amplified as controls. The Chop-PCR primers are listed in the Supplementary Data 7.

For RT-qPCR, total RNA of the same material was extracted using RNA-easy Isolation Reagent (Vazyme #R701-01). As ELE-RNA may not be polyadenylated, the RNA was then treated using First Strand Synthesis Kit (Vazyme #R312) with random hexamers and oligo(dT) 20VN Primers for reverse transcription. The expression levels of individual ELE-RNAs were quantified by qPCR using SYBR green master mix (Vazyme #Q111) on the BIO-RAD CFX96 real-time system according to established protocols. *TaActin (TraesCS1A01G274400)* was used

as an internal control to normalize the expression levels of the ELEs. Three biological replicates were applied for each ELE. Primers are listed in Supplementary Table 5.

## Reporting summary

Further information on research design is available in the Nature Portfolio Reporting Summary linked to this article.

## Data availability

The datasets generated during the current study include CAGE-seq, RNA-seq, ChIP-seq, Bisulfite-seq, smRNA-seq and ALE-seq are available in the Gene Expression Omnibus (GEO) repository under accession GSE198284. Other datasets including Chinese Spring Hi-C data[9], RNA-seq, ChIP-seq and Bisulfite-seq of Chinese Spring seedlings[26], nascent RNA sequencing[17], RNA-seq of Chinese Spring tissues during development and under treatments[27] were published previously. Source data are provided with this paper.

## Code availability

Scripts are available at Github [https://github.com/yilinlinyi/wheat_ELE_spike].

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

## Acknowledgements

This study was supported by the National Natural Science Foundation of China (31921005 XYB, 32270628 ZYJ and 32101739 ZJF), the National Science Fund for Excellent Young Scholars (32022012 ZYJ) and State Key Laboratory of Plant Cell and Chromosome Engineering, Institute of Genetics and Developmental Biology (PCCE-KF-2023-03 ZYJ). We thank Dr. Deng-Cai Liu from Sichuan Agricultural University and Dr. Lei Gong from Northeast Normal University for helpful comments.

## Author contributions

Y.J.Z., Y.B.X., Z.C.D. and J.C. conceived and designed the experiments. Z.J.L., S.B.Y., Y.E.Z., J.F.Z., H.S.D., L.H.Y., Y.L.Z., W.T., W.L.Z. and Y.P.T. performed the experiments. Y.L.X., J.Y.Z., M.Y.W., H.Y.W., Y.Y.Z., F.Z. and Y.J.Z. analyzed the data. Y.J.Z. wrote the manuscript with input from all authors.

## Competing interests

The authors declare no conflict of interest.
