## [Peer Review File · Nature Communications]

Transposable element-initiated enhancer-like elements generate the subgenome-biased spike specificity of polyploid wheatReviewers' Comments:

Reviewer #1:

Remarks to the Author:

In this article authors identified loci producing putative active enhancer transcripts (eRNAs). By integrating data from gene expression cap analysis and epigenome profiling at different developmental stages authors quantified the expression of the putative eRNA. Authors found that eRNAs are mainly initiated from RLG_famc7.3. Authors observed that Knockdown of RLG_famc7.3-initiated eRNAs and resulted in a global downregulation of spikelet-specific genes and abnormal spikelet development. Overall, this study could be potentially interesting. However, I have important concerns. The study is mainly based on correlation and because correlation does not mean causality I am not convince that authors demonstrated that

(i) eRNA are produced from RLG_famc7.3. What is the definition of eRNA? Why not calling them just non coding RNA? Where is the demonstration that RLG_famc7.3 is an enhancer? Why not hypothesizing that RLG_famc7.3 produce a coding mRNA that impact spikelet-specific genes? How much confident we are that the RNA produce is really non coding?

(ii) I do not see any experience proving that the RNA generated by the RLG_famc7.3 locus control directly the spikelet-specific genes.

In addition the logic of manuscript is difficult to follow.

Reviewer #2:

Remarks to the Author:

In this manuscript Xie and collaborators analysed wheat transcripts produced from expression of enhancer elements (eRNAs) and found that a large proportion originated from a transposable element family which specifically expanded in the subgenome A (wheat has a polyploid genome). Further, authors demonstrated that interfering with expression of these TE-derived eRNA has physiological consequences in spikelet development and affect gene expression of spikelet specific genes. The dynamic of eRNAs expression is still poorly investigated in plants, and this work uses a combination of cutting-edge genomics approaches to characterise these molecules in an economically important crop. In addition, this manuscript highlights an important regulatory role of TEs in wheat which could be sub-genome specific. For all these reasons, I believe this work break a new ground in the study of genome plasticity in plants and will very likely attract a high level of attention in the scientific community.

I have a series of comments for authors, which I believe could be considered to improve the manuscript.

There is a general excessive parsimony in introducing critical concepts, background, terminology, and analyses applied, which collectively make difficult to follow the text. For examples, the RLG_famc7.3 TE family is not properly introduced (TEs are grouped in classes and superfamilies with significant different genetic properties). Also, there is very little information provided concerning analyses like the hidden Markov model (e.g. parameters used, training dataset) or the TF binding search. Related to the TF binding analysis, from Figure 3g it appears that BPC1 and BPC5 are more present in RLG_famc7.3 rather than BPC6 and RAMOSA1 which are displayed in the figure. What about the other two? Are also enriched in AG? In addition, both BPC6 and RAMOSA1 appear to actually be just AG repeats, is this maybe linked to the genetic structures of the RLG_famc family (e.g. LTR domain)? Concerning the RNAi lines, is there any proof that RLG_famc7.3 expression have been suppressed e.g. any gels/assays to show that RLG_famc7.3 is knocked down in the RNAi lines?

Importantly, few figure legends have been cut so that the full text is not available to review.

Therefore, it is not possible to fully evaluate the latest panels in these figures.

Remarkably, the discussion section is very short and doesn't touch upon most of the results and the significance of the findings. IT would be ideal to expand this part analysing for examples if RLG_fam7.3 is present in landraces? OR evidence for similar in other plant species or organisms? The evolutionary advantage of this having occurred? Which eRNA targets could cause this? Etc...

It also appears that two different wheat varieties have been used for the genomics analysis and the transgenic experiments (I think this is a normal approach because Fielder is the main genotype used for transformation experiment). However, all material used (including Fielder genotype) should be described in the appropriate "Material" section in the manuscript.

Other minor points:

Figure 1e. What is the meaning of the negative values (0, -1) in the DNA methylation track profiles? I was assuming this was methylation in the negative strand, but if this is the case it looks quite weird (i.e. CG methylation appear to be strand specific in large area of the seedling track??)

Line 103- maybe a reference is missing?

Lines 131-136- very long sentence, revise

Figure 1g in the legend the description of regions does not match what is in the figure

Fig 2a-b- It should be explained in legend how the genes and enhancers were ordered in the heatmaps. In each enhancer matching vertically the corresponding gene?

Fig 2d- not entirely clear what it is showing, could the legend be more specific?

Fig4c-f- how many plants or spikes were measured?

Line 257-258- is there an impact on root growth and expression profiles grown in Hoagland's vs soil? There is any previous work showing lack of difference or if there's an impact on the rest of the plant too?

Line 331, 334, 336, 338, 421, 422- no references for the software used.

Reviewer #3:

The manuscript Transposable element-initiated enhancer RNAs generate the subgenome-biased spikelet specificity of polyploid wheat by Xie et al. address the role of transposable elements in gene regulation of specific sub-genomes in hexaploidy wheat.

The study describes the generation of a rich omic's dataset, including CAGE-seq, the use of these resources to identify TE-associated enhancer transcripts (eRNAs), characterization of the sub-genome specific regulation of the identified TE-enhancers, and finally functional validation through phenotypic characterization of knock-down lines of eRNAs from one subclass of these TE-enhancers.

The idea behind the study is undoubtedly very interesting - probing the mechanistic basis for sub-genome specific contribution to polyploid wheat biology. Unfortunately, I find that the current version of the manuscript has major shortcomings related to the justification of methods/approaches, clarity of methodological details, the presentation of results (or lack thereof).

Introduction:

1. The introduction would provide a better background for the paper if the authors added some more background to make it easier for the general readership to understand the link between eRNA transcription and enhancer function through enhancer-promoter contacts/looping structures.
2. The first paragraph of the Intro is particularly unclear, including many misleading statements.
 - Lines 59-61: “..wheat merged three sets..” sounds like someone actively merged these sub-genomes. Should rather be phrased as “wheat genome consists of”
 - Line 66: What is mean by “spatiotemporal specificity” ? Of what?
 - Lines 62-64: What is meant by the description of genome evolution as being “almost turned over”?
 - Line 67-68: Genetic drift is one of the evolutionary forces, but “regulatory drift” is not a thing as far as I know. Please rephrase this sentence.

Results and discussion:

- General clarity and precision of the presentation of the results could be improved. Many of the important results have been buried in the extensive supplementary notes and many concluding statements about genomic patterns are not backed up by details about the type of statistic test and/or p-value, effect size etc. Some examples follow:
 - o Line 144: “largely correlated”

- Lines 152-155/Figure 2d: It is unclear what is compared in these analyses and also unclear how the observed pattern leads to the authors' conclusions.
 - Line 162: The observation that 20% of eRNAs comes from TE-associated DNA is used to back up a general statement that TEs can be a “rich source” of new cis-regulatory elements. When >80% of the total wheat DNA comes from TEs, 20% eRNA from TEs seems low to me. Perhaps the authors could test if TE-eRNAs are underrepresented compared to what is expected if TE-enhancers evolve by chance?
 - Lines 202-203: what kind of test, and p-value?
 - Line 166-167: “predominantly detected in”, please quantify and report numbers used to draw these conclusions.
 - Line 174: “exhibited the most obvious tissue specificity” – is this possible to quantify in some ways? One way could be to calculate some tissue-specificity index (e.g. TAU) and then do a more formal analysis of the distribution of tissue-specificity across TE-eRNAs in general?
- What is the justification for using bisulfite sequencing to annotate TE enhancers?
 - The title of the paper and start of the results section (lines 96-141) communicates that the study focuses on the genomic landscape of TE-enhancer transcription. However, the authors choose to present very few in-depth analyses of TEs-eRNAs. The TE focus seems to appear in the results section from line 159, but now the authors have decided to only focus on a single sub-class of TE-eRNAs without any more justification than that this is the biggest single contributor to TE-eRNAs (22%, which leaves 78% of TE-eRNAs discarded from being analyzed in any more depth). The authors should consider expanding the analyses of TE-derived eRNAs in the main text to include a genome-wide overview of which superfamilies/families of TEs contribute to the TE-eRNA repertoire. It would also be interesting to know if any taxonomic classes are under/overrepresented as sources for eRNA.
 - The results in the section starting on line 142 “*CAGE identifies subgenome-partitioned tissue-specific eRNA TSS*” is based on a statistical method the authors have developed. It would be nice if the authors could help the non-expert readership to understand the essence of this method— what does the method capture? When does a gene get assigned many vs few enhancers? The authors should also try to describe the results a bit more: Did genes in sub-genomes have similar numbers of enhancers? Were the enhancer numbers different across different tissues, and perhaps most importantly, how were TE-enhancers distributed across the subgenomes/tissues?

- The authors describe the results in figure 2b (line 149) as *'The tissue specificity of eRNAs and target genes were highly consistent (Fig. 2b–c).'* Please describe better what is meant by “consistent”.
- Several places in the manuscript, the authors make ‘big’ conclusions. Unfortunately, it’s not always easy to understand the logic behind all these conclusions, and in some instances, the authors should also consider presenting different interpretations of a results. Here are some examples:
 - *“Here, we detected a cohort of TE-initiated enhancer RNAs specifically expanded in subgenome A, and demonstrated the direct impact on regulating spikelet specificity, which bridges the mechanistic gaps between TE bursts, regulatory specificity, and polyploid developmental plasticity (Fig. 5)”* Exactly how does the results in the paper help “bridge a mechanistic gap” between TE evolution and developmental plasticity? Please expand with a sentence or two to make it easy for the reader to follow the logic.
 - *“Almost no eclDNA was detected for RLG_famc7.3 (Table S1). Instead, the tissue specificity of eRNAs was highly consistent with that of the target genes (Fig. 2a–b). These results suggest that the TE embedded enhancers are co-opted for the spatiotemporal regulation of host genes.”* I agree that the lack of eclDNA is strong evidence for these TEs not being active, however, this does not mean that they are (all of them) co-opted for wheat gene regulation. Some might, yes, but others eRNA-loci might also just be transcribed as a collateral effect of these sequences harbour motifs that bind host-regulatory proteins (evolving under drift). Highlighting such nuances would benefit this paper.
 - *“Distinct and tissue-related functional enrichment was detected among the genes targeted by tissue-specific eRNAs (Fig. S15), confirming the functional relevance of eRNAs for developmental specificity.”* If I understand the analyses correct, the eRNAs have been linked to a gene based on (at least partially) the correlation between eRNA expression and gene expression. If this is correct, I wonder if this conclusion is a bit circular? Gene expression correlates with chromatin accessibility in regions flanking the genes. Wouldn’t we thus expect correlated transcription of non-protein coding RNA, such as eRNA, and genes as a consequence? If the answer to this is yes, then doesn’t these analyses just

confirm that tissue-specific genes are enriched for functions relevant for these tissues? I have a hard time understanding the jump to “eRNA functional relevance” from these results.

We sincerely thank the reviewers for the positive evaluation and cogent comments to help us improve our manuscript, which we have addressed in detail below.

Reviewer #1 (Remarks to the Author):

In this article authors identified loci producing putative active enhancer transcripts (eRNAs). By integrating data from gene expression cap analysis and epigenome profiling at different developmental stages authors quantified the expression of the putative eRNA. Authors found that eRNAs are mainly initiated from RLG_famc7.3. Authors observed that Knockdown of RLG_famc7.3-initiated eRNAs and resulted in a global downregulation of spikelet-specific genes and abnormal spikelet development. Overall, this study could be potentially interesting. However, I have important concerns. The study is mainly based on correlation and because correlation does not mean causality, I am not convince that authors demonstrated that

1. eRNA are produced from RLG_famc7.3. What is the definition of eRNA? Why not calling them just non coding RNA? Where is the demonstration that RLG_famc7.3 is an enhancer? Why not hypothesizing that RLG_famc7.3 produce a coding mRNA that impact spikelet-specific genes? How much confident we are that the RNA produce is really non coding?

Response: We thank the reviewer for raising a series of fundamental questions regarding the underlying concepts and definitions in our manuscript. After careful consideration, we have systematically sorted out the logic and writing. The description and conclusions in the revised version are now more rigorous. The point-to-point response is as follows.

1) Definition:

The definition of enhancer RNA follows the way of eRNA definition in animals, which are non-coding transcripts produced by enhancers. Enhancer elements are detected based on the combination of epigenetic marks typically present in enhancers. However, although the use of epigenomic map is a popular strategy for defining genome-wide enhancers (distal regulatory elements of genes), the presence of active epigenetic marks suggests a regulatory potential of enhancers rather than a validated function. To be more rigorous, we defined gene-distal regulatory elements as enhancer-like elements (ELEs), and the produced RNA was defined as ELE-RNA instead of eRNA in the revised manuscript. Please refer to lines 48-50, lines 77-93.

2) Detection:

Transcription of eRNAs occurs very early in the gene transcription process, which is primarily detected by nascent RNA detection strategies, including GRO-seq and CAGE. In our dataset, we analyzed the capped transcription start sites of transcripts by CAGE-seq to detect nascent RNAs, which showed higher abundance compared to mature mRNAs. Detected nascent transcripts with significant coding potential are removed. Thus, the defined ELE-RNAs are less abundant in mature RNA and less likely

to have coding products. Please refer to Fig.1h, lines 94-99 and lines 124-140.

3) Conclusion:

This manuscript predicts specific effects of TE-derived ELEs on wheat spike development by analyzing the transcriptome of nascent RNAs produced by ELEs, which was further validated by experiments. We agree that we cannot distinguish the effect of RLG_famc7.3-embedded ELE or ELE-RNA on spike development. Given that the function of distal regulatory elements (enhancer-like elements, ELEs) is difficult to be determined from sequence, active transcription serves as a good marker for the spatiotemporal function of distal REs. These findings detect, for the first time, tissue-specific expression of TE-derived nascent non-coding transcripts in plants and provide genetic evidence supporting a spike-specific function of TE-embedded regulatory elements. Given the high abundance of TEs in common wheat, both strategy and conclusions provide useful clues for elucidating the subgenome-specific functions of TEs and their impact on polyploid plasticity. Please refer to lines 241-247 and lines 272-292.

2. I do not see any experience proving that the RNA generated by the RLG_famc7.3 locus control directly the spikelet-specific genes.

Response: We thank the reviewer for pointing out this issue. From the current results, we detected the impact of RLG_famc7.3 locus on spike development. Despite that we can't distinguish whether the effects are directly influenced by RLG_famc7.3-embedded ELEs or ELE-RNAs, the present study highlights the strategy of employing eRNAs to predict enhancer function, which is further validated by experiments. Assessing enhancer function is challenged by the varying distance between enhancers and targets, and the multiple correspondences between enhancers and targets. Using eRNA to predict enhancer function bypass the step of detecting targets. In the revised manuscript, we provided a list of potential targets of RLG_famc7.3-derived ELE-RNAs (lines 226-229 and Table. S7), which have correlated expression with RLG_famc7.3 and reduced expression in ELE-RNA knockdown lines.

3. In addition the logic of manuscript is difficult to follow.

Response: We carefully reorganized the logic and writing of the manuscript. We first detected a large group of TE-derived regulatory elements distant to coding genes, namely, enhancer like elements (ELEs). By profiling the noncoding nascent transcripts across tissues, we found that 20% of these transcripts are derived from TEs, most of which were specifically expressed in spikes and associated with specific TE expansion in diploid progenitor of subgenome A. Further knockdown of ELE-RNAs leads to aberrant spike development, implying specific developmental regulation of ELEs. The strategy for predicting ELE functions and the findings provide insightful clues for elucidating the function of TEs, which account for a large proportion of common wheat genome.

Reviewer #2 (Remarks to the Author):

In this manuscript Xie and collaborators analysed wheat transcripts produced from expression of enhancer elements (eRNAs) and found that a large proportion originated from a transposable element family which specifically expanded in the subgenome A (wheat has a polyploid genome). Further, authors demonstrated that interfering with expression of these TE-derived eRNA has physiological consequences in spikelet development and affect gene expression of spikelet specific genes. The dynamic of eRNAs expression is still poorly investigated in plants, and this work uses a combination of cutting-edge genomics approaches to characterise these molecules in an economically important crop. In addition, this manuscript highlights an important regulatory role of TEs in wheat which could be sub-genome specific. For all these reasons, I believe this work break a new ground in the study of genome plasticity in plants and will very likely attract a high level of attention in the scientific community. I have a series of comments for authors, which I believe could be considered to improve the manuscript.

Response: We thank the reviewer for the positive comments on our work and appreciate all the comments and suggestions for improving the manuscript.

1. There is a general excessive parsimony in introducing critical concepts, background, terminology, and analyses applied, which collectively make difficult to follow the text. For examples, the RLG_famc7.3 TE family is not properly introduced (TEs are grouped in classes and superfamilies with significant different genetic proprieties). Also, there is very little information provided concerning analyses like the hidden Markov model (e.g. parameters used, training dataset) or the TF binding search. Related to the TF binding analysis, from Figure 3g it appears that BPC1 and BPC5 are more present in RLG_famc7.3 rather than BPC6 and RAMOSA1 which are displayed in the figure. What about the other two? Are also enriched in AG? In addition, both BPC6 and RAMOSA1 appear to actually be just AG repeats, is this maybe linked to the genetic structures of the RLG_famc family (e.g. LTR domain)?

Response: We added detailed descriptions according to the suggestions.

- 1) We added detailed description to the definition and detection of ELE-RNAs (lines 75-93).
- 2) We added details about the detection of TE families (see Methods lines 465-478) and the analysis and description of TE-derived ELE-RNAs (Fig.3a-b and lines 179-184, lines 192-199)
- 3) We added a more thorough explanation of the hidden Markov model in Methods (lines 371-376).
- 4) We added details of the TF binding search in Methods (lines 424-426).
- 5) BCP1,5,6 and RAMOSA1 are all AG-rich motifs. We only presented two logos in Fig. 3g due to space constraints. We now changed these two logos to BPC1 and RAMOSA1 (Fig. 3f).
- 6) We compared the distributions of AGAG repeats and AG-rich motifs along the full-

length RLG_famc7.3 (the following figure). Four AG-rich motifs were more abundant in the border regions of LTRs, while AG repeats were randomly distributed along RLG_famc7.3.

Distribution of AGAG repeats and AG-rich motifs along RLG_famc7.3.

- Sequence logos of the four motifs BPC1, BPC5, BPC6 and RAMOSA1.
- “AGAG” sequence frequency along the full-length LTR. The RLG_famc7.3 and other top abundant TE families.
- The distribution of the four AG-rich motifs along the full-length RLG_famc7.3.

2. Concerning the RNAi lines, is there any proof that RLG_famc7.3 expression have been suppressed e.g. any gels/assays to show that RLG_famc7.3 is knocked down in

the RNAi lines?

Response: We thank the reviewer for pointing out this issue. Enhancer RNAs are generally detected based on nascent RNA sequencing methods. RNAi occurs at post-transcriptional level. To examine the effect of RNAi, we performed small RNA sequencing and detected a significant accumulation of small RNAs surrounding RLG_famc7.3-initiated eRNAs (Fig. S19 and lines 218-219).

3. Importantly, few figure legends have been cut so that the full text is not available to review. Therefore, it is not possible to fully evaluate the latest panels in these figures. Remarkably, the discussion section is very short and doesn't touch upon most of the results and the significance of the findings. It would be ideal to expand this part analysing for examples if RLG_fam7.3 is present in landraces? OR evidence for similar in other plant species or organisms? The evolutionary advantage of this having occurred? Which eRNA targets could cause this? Etc...

Response: We thank the reviewer for pointing out these issues.

- 1) We adjusted the figure to show the full legend.
- 2) We enriched the Discussion section about the evolution of TE-derived ELEs (lines 248-259). Compared to other TE-derived ELEs, RLG_famc7.3-derived ELEs are less conserved between diploid progenitor and hexaploid common wheat (Fig. S22a) and are largely conserved across hexaploid and tetraploid wheat (Fig. S22b), indicating this TE family may have undergone evolutionary selection following hexaploidization, and were fixed in polyploid wheat.
- 3) The expansion and evolution of RLG_famc7.3 are assessed in the main text (lines 197-199, Fig. 3c and Fig. S18).
4. It also appears that two different wheat varieties have been used for the genomics analysis and the transgenic experiments (I think this is a normal approach because Fielder is the main genotype used for transformation experiment). However, all material used (including Fielder genotype) should be described in the appropriate "Material" section in the manuscript.

Response: We thank the reviewer for pointing out this issue. Data were generated in Chinese Spring and the transgenic experiments were performed in JW1. Before experimental validation, we systematically compared the sequences and epigenetic profiles of ELEs between these two cultivars. For the ELEs with conserved sequences between the two cultivars, the epigenetic activity is also highly consistent. Additionally, 352 of 411 spike active RLG_famc7.3 enhancer like elements are also had active chromatin state in JW1 spike, indicating RLG_famc7.3-derived ELEs already present in the common ancestor of these cultivars (Fig.S25).

Other minor points:

1. Figure 1e. What is the meaning of the negative values (0, -1) in the DNA methylation track profiles? I was assuming this was methylation in the negative strand, but if this is the case it looks quite weird (i.e. CG methylation appear to be strand specific in large area of the seedling track??)

Response: We thank the reviewer for the thorough evaluation of our manuscript. Negative values in the data indicate negative strands. We carefully examined the data and found that this phenomenon is common in seedling data but not in other tissues. Since the seedling methylation data were obtained from public resource (Methods, lines 339-341 and line 406), while other tissues were generated by our lab, we are unclear about the reasons behind this peculiar phenomenon. To prevent any potential misinterpretation, we replaced the genome browser view with data from other tissues we generated and did not use public methylation data for analysis.

2. Line 103- maybe a reference is missing?

Response: We thank the reviewer for pointing out the issue. We revised the previous statement “Enhancers are essential for spatiotemporal specificity” to “Recent human studies demonstrated prevalent and early production of noncoding transcripts from active enhancers, which are excellent markers for directly predicting enhancer functions in developmental specificity”, and added appropriate references. Please refer to lines 120-123.

3. Lines 131-136- very long sentence, revise

Response: We thank the reviewer for the suggestions. We split this sentence into two sentences. Please refer to lines 156-161.

4. Figure 1g in the legend the description of regions does not match what is in the figure.

Response: We thank the reviewer for pointing out the error. We revised the legend of Figure 1g.

5. Fig 2a-b- It should be explained in legend how the genes and enhancers were ordered in the heatmaps. In each enhancer matching vertically the corresponding gene?

Response: We thank the reviewer for the suggestions. Fig.2b lists the predicted targets of ELEs shown in Fig 2a, based on correlated expression between ELE-RNA and nearby genes. Since one ELE may have multiple targets and one gene may be targeted by multiple ELEs, there is not a one-to-one correspondence between ELEs and targets. Genes in Fig. 2b are ordered according to their expression levels in each tissue.

6. Fig 2d- not entirely clear what it is showing, could the legend be more specific? Fig4c-f- how many plants or spikes were measured?

Response: We thank the reviewer for the suggestions. We revised the legends of Fig 2d and add an illustration below the figure. In Fig4c-f, 27 spikes were used to measure the phenotype. The specific information of each spike refers to Table S12.

7. Lines 257-258- is there an impact on root growth and expression profiles grown in Hoagland's vs soil? There is any previous work showing lack of difference or if

there's an impact on the rest of the plant too?

Response: To prepare the root samples for sequencing, the plants were grown in Hoagland solution because the soil was difficult to clean. We downloaded public data including root transcriptome samples grown in soil (GSM5910155), which showed a high expression correlation with our root data.

8. Line 331, 334, 336, 338, 421, 422- no references for the software used.

Response: We added the references for each software at lines 346, 392, 394, 396, 494. There's no manuscript for TransDecoder, and we cited the github link (<https://github.com/TransDecoder/TransDecoder>).

Reviewer #3 (Remarks to the Author):

The manuscript Transposable element-initiated enhancer RNAs generate the subgenome- biased spikelet specificity of polyploid wheat by Xie et al. address the role of transposable elements in gene regulation of specific sub-genomes in hexaploidy wheat.

The study describes the generation of a rich omic's dataset, including CAGE-seq, the use of these resources to identify TE-associated enhancer transcripts (eRNAs), characterization of the sub-genome specific regulation of the identified TE-enhancers, and finally functional validation through phenotypic characterization of knock-down lines of eRNAs from one sub- class of these TE-enhancers.

The idea behind the study is undoubtedly very interesting - probing the mechanistic basis for sub-genome specific contribution to polyploid wheat biology. Unfortunately, I find that the current version of the manuscript has major shortcomings related to the justification of methods/approaches, clarity of methodological details, the presentation of results (or lack thereof).

Response: We thank the reviewer for the positive comments on our work. We have added more detailed descriptions based on the reviewer's suggestions. Please refer to the sentences in blue in the revised manuscript, and point-to-point responses are as follows.

Introduction:

1. The introduction would provide a better background for the paper if the authors added some more background to make it easier for the general readership to understand the link between eRNA transcription and enhancer function through enhancer-promoter contacts/looping structures.

Response: We added a paragraph to describe the background and definition of eRNA. To be more rigorous, we defined gene-distal regulatory elements as enhancer-like elements (ELEs), and the produced RNA was defined as ELE-RNA instead of eRNA in the revised manuscript. Please refer to lines 48-50, lines 77-93.

2. The first paragraph of the Intro is particularly unclear, including many misleading statements.

Response: We thank the reviewer for the thorough reading and comments on our manuscript. We have revised the first paragraph of the Introduction to make it clearer.

- 1) Lines 59-61: “..wheat merged three sets..” sounds like someone actively merged these sub-genomes. Should rather be phrased as “wheat genome consists of”

Response: We revised the statement as suggested, please refer to lines 61-62.

- 2) Line 66: What is mean by “spatiotemporal specificity” ? Of what?

Response: We revised the sentence to “However, a causal relationship between TE-

embedded REs and polyploid plasticity has so far been missing”. Please refer to lines 68-70.

- 3) Lines 62-64: What is meant by the description of genome evolution as being “almost turned over”?

Response: We revised “almost turned over” to “highly divergent”, please refer to line 65.

- 4) Lines 67-68: Genetic drift is one of the evolutionary forces, but “regulatory drift” is not a thing as far as I know. Please rephrase this sentence.

Response: We rephrased this sentence. Please refer to lines 70-71

Results and discussion:

3. - General clarity and precision of the presentation of the results could be improved. Many of the important results have been buried in the extensive supplementary notes and many concluding statements about genomic patterns are not backed up by details about the type of statistic test and/or p-value, effect size etc. Some examples follow:

- 1) Line 144: “largely correlated”

Response: We added detailed statistics for this statement, “ELE-RNAs are largely correlated with nearby gene expression levels” (line 166 in the revised version), the statistic test is Welch Two Sample t-test, and with the p-value < 2.2e-16. Please refer to the legend of Fig S14.

- 2) Lines 152-155/Figure 2d: It is unclear what is compared in these analyses and also unclear how the observed pattern leads to the authors conclusions.

Response: We revised the legend and add an illustration below the figure and more details in the methods section “ELE comparison across subgenomes” (lines 452-455). This analysis was conducted to evaluate the transcriptional activity of tissue-specific and ubiquitously expressed enhancers in other subgenomes. First, we identified homologous enhancers across subgenomes, and for each enhancer transcript, we examined the expression of homologous enhancers. Compared with the ubiquitously expressed ELE-RNAs, the sequences of tissue-specifically expressed ELE-RNAs were less conserved across subgenomes (Fig. 2d), indicating a proportion of tissue-specific functions of ELEs are independently evolved across subgenomes.

- 3) Line 162: The observation that 20% of eRNAs comes from TE-associated DNA is used to back up a general statement that TEs can be a “rich source” of new cis-regulatory elements. When >80% of the total wheat DNA comes from TEs, 20% eRNA from TEs seems low to me. Perhaps the authors could test if TE-eRNAs are underrepresented compared to what is expected if TE-enhancers evolves by chance?

Response: We detected 20% of enhancer transcription is initiated by TE, which is greater than the proportion of gene transcription initiated by TE (4.4%). We agree with the reviewer that the majority of TE-derived enhancers may have arisen from genetic drift, as there is no enrichment of TE-generated ELE-RNAs as a whole compared to the genomic background. However, among the most abundant TEs in the genome, RLG_famc7.3 apparently enriched for ELE-RNAs (Fig. 3b and lines 193-199), implying that the evolutionary selection and fixation of this TE family may not be due

to random drift.

4) Lines 202-203: what kind of test, and p-value?

Response: “no significant difference was detected in vegetative growth” was to “no apparent difference was observed in vegetative growth ” between RNAi and WT lines (line 218).

5) Line 166-167: “predominantly detected in”, please quantify and report numbers used to draw these conclusions.

Response: We provide the number of each TE family in each subgenome in the supplemental Tabs S5.

6) Line 174: “exhibited the most obvious tissue specificity” – is this possible to quantify in some ways? One way could be to calculate some tissue- specificity index (e.g. TAU) and then do a more formal analyses of the distribution of tissue-specificity across TE-eRNAs in general?

Response: We describe more formally the statistical analysis of the tissue-specificity of TE-derived eRNAs. Fisher's exact test was used to test the enrichment of TE-eRNA in tissue-specific eRNA. Please refer to the description of Fig. 3d for details.

4. What is the justification for using bisulfite sequencing to annotate TE enhancers?

Response: We apologize for the unclear description of TE enhancer annotation. Enhancers are defined based on the combination of typical epigenetic profiles detected in plant enhancers as previously reported (doi: 10.1186/s13059-019-1746-8), including enrichment of H3K9ac and H3K4me3, and depletion of DNA methylation. Please refer to lines 147-152.

5. The title of the paper and start of the results section (lines 96-141) communicates that the study focusses on the genomic landscape of TE-enhancer transcription. However, the authors choose to present very few in-depth analyses of TE-eRNAs. The TE focus seems to appear in the results section from line 159, but now the authors have decided to only focus on a single sub-class of TE-eRNAs without any more justification than that this is the biggest single contributor to TE-eRNAs (22%, which leaves 78% of TE-eRNAs discarded from being analyzed in any more depth). The authors should consider expanding the analyses of TE-derived eRNAs in the main text to include a genome wide overview of which superfamilies/families of TEs that contributes to the TE-eRNA repertoire. It would also be interesting to know if any taxonomic classes are under/overrepresented as sources for eRNA.

Response: We thank the reviewer for the suggestion. We expanded the analysis of TE-derived eRNAs in the manuscript to include a genome-wide overview of the types and families of TEs contributed to the TE-derived eRNA repertoire, and the under/overrepresented families producing eRNAs. Please refer to lines 192-199 and Figure 3 for details. In addition, we revised the manuscript to make the logic clearer, including the detection of distant regulatory elements (ELEs), prediction of ELE functions by profiling nascent transcripts associated with lineage-specific TE expansion, and experimental validation.

6. The results in the section starting on line 142 “CAGE identifies subgenome-partitioned tissue-specific eRNA TSS” is based on a statistical method the authors have developed. It would be nice if the authors could help the non-expert readership to understand the essence of this method– what does method capture? When does a gene get assigned many vs few enhancers? The authors should also try to describe the results a bit more: Did genes in sub-genomes have similar numbers of enhancers? Were the enhancer numbers different across different tissues, and perhaps most importantly, how was TE-enhancers distributed across the subgenomes/tissues?

Response: We thank the reviewer for raising these issues. We added more analysis of eRNA and TE-derived eRNA distribution in the section “CAGE identifies subgenome-partitioned tissue-specific eRNA TSS”, which is now changed to “CAGE identifies subgenome-partitioned tissue-specific transcription from TE-embedded ELEs.

- 1) Target definition. Putative targets of enhancers were assigned by employing previously proposed strategies (doi:10.1093/plcell/koab028, doi: 10.1016/j.molp.2022.12.018, doi: 10.1016/j.molp.2022.12.018) that combines the distance of regulatory elements (REs) to genes and the correlation between epigenetic density of REs and expression of nearby genes. Here we show that the expression of REs is more correlated with nearby genes than the epigenetic state of REs, and further integrated the distance between REs and genes and expression correlations between RE-RNA and genes to define enhancer targets (Fig.S13b, lines 166-170, lines 435-444).
- 2) Distribution and tissue specificity of ELEs and ELE-RNAs. Comparison across three subgenomes suggested that subgenome A contributed more abundant TE-derived ELEs and nascent transcripts (Fig. S15 and lines 183-184). In addition, we added more description about the contribution of TEs from different families to ELEs and ELE-RNAs (Fig. 3).

7. The authors describe the results in figure 2b (line 149) as ‘The tissue specificity of eRNAs and target genes were highly consistent (Fig. 2b–c).’ Please describe better what is meant by “consistent”.

Response: We thank the reviewer for this suggestion. We revised the statement accordingly. “The putative ELE-RNA targets were thus defined by integrating information from both gene proximity and expression correlation (please refer to the Methods), resulting in a list of putative targets with consistent tissue specificity with corresponding ELE-RNAs (Fig. 2b–c)” (lines 166-170).

8. Several places in the manuscript, the authors make ‘big’ conclusions.

Unfortunately, it’s not always easy to understand the logic behind all these conclusions, and in some instances, the authors should also consider presenting different interpretations of a results. Here are some examples:

- 1) “Here, we detected a cohort of TE-initiated enhancer RNAs specifically expanded in subgenome A, and demonstrated the direct impact on regulating spikelet specificity, which bridges the mechanistic gaps between TE bursts, regulatory

specificity, and polyploid developmental plasticity (Fig. 5)” Exactly how does the results in the paper help “bridge a mechanistic gap” between TE evolution and developmental plasticity? Please expand with a sentence or two to make it easy for the reader to follow the logic.

Response: We thank the reviewer for pointing this issue out. We revised the conclusions of the manuscripts to be more rigorous. The previous conclusion about ‘bridge a mechanistic gap’ is now revised to “The strategy and findings help to elucidate the causal effects of TEs on agronomic traits, providing insight into the direct regulatory function of numerous TEs in common wheat, and their contribution to developmental specificity and polyploid plasticity (Fig. 5)”. Please refer to lines 241-247.

2) “Almost no eDNA was detected for RLG_famc7.3 (Table S1). Instead, the tissue specificity of eRNAs was highly consistent with that of the target genes (Fig. 2a–b). These results suggest that the TE embedded enhancers are co-opted for the spatiotemporal regulation of host genes.” I agree that the lack of eDNA is strong evidence for these TEs not being active, however, this does not mean that they are (all of them) co-opted for wheat gene regulation. Some might, yes, but other eRNA-loci might also just be transcribed as a collateral effect of these sequences harbour motifs that bind host-regulatory proteins (evolving under drift). Highlighting such nuances would benefit this paper.

Response: We thank the reviewer for pointing out this issue. We revised the conclusion to “Thus, it is likely that a subset of TEs generated active enhancers and mediated reciprocal adaptation between TEs and hosts”. Please refer to lines 270-271.

3) Distinct and tissue-related functional enrichment was detected among the genes targeted by tissue-specific eRNAs (Fig. S15), confirming the functional relevance of eRNAs for developmental specificity.” If I understand the analyses correct, the eRNAs have been linked to a gene based on (at least partially) the correlation between eRNA expression and gene expression. If this is correct, I wonder if this conclusion is a bit circular? Gene expression correlates with chromatin accessibility in regions flanking the genes. Wouldn’t we thus expect correlated transcription of non-protein coding RNA, such as eRNA, and genes as a consequence? If the answer to this is yes, then doesn’t these analyses just confirm that tissue-specific genes are enriched for functions relevant for these tissues? I have a hard time understanding the jump to “eRNA functional relevance” from these results.

Response: We agree with the reviewer’s comments. In the revised manuscript, we have removed this conclusion, and added details regarding the definition of putative targets, the statistics of enhancers, and eRNA distribution across subgenomes and tissues as suggested by the reviewer’s above comments. Please refer to lines 166-170 and Figs S15 for details.

Reviewers' Comments:

Reviewer #1:

Remarks to the Author:

In this revised version authors improved their manuscript.

Reviewer #2:

Remarks to the Author:

This is a revised manuscript from Xie and co-workers, in which authors implemented many of my previous comments and produced an improved version, enhancing the text clarity and providing a better explanation for their analyses. There are only two points which remain outstanding, which are: 1) With the new data provided it appears that AG-rich TF binding enrichment is always located upstream of expression on the RLG_famc7.3 family, and this appears to be a structural domain in LTR. This is very interesting, but I was wondering if authors have checked if these AG-rich sequences in the entire wheat genome (not associated to TEs) are associated to generation of ELE RNAs ? In other words, can AG-motifs be considered general promoters for ELE RNAs in wheat, or the association with the TE is important for their expression/function ?

2) Figure S19 indeed displays production of sRNAs in knockdown lines. To be informative, I guess an additional track with sRNAseq from a control wild-type non-transformed line should also be presented (showing absence of siRNA accumulation at the same location). Moreover, I guess that in absence of a CAGEseq or RNAseq data performed on these knockdown lines (or by different molecular approach e.g. RT-qPCR), authors cannot assume that an increased amount of siRNA will reflect a real knockdown of the corresponding target Enhancer RNA. I believe in absence of these data, in the manuscript should be mentioned that the suppression of the target, however probable, has not been investigated directly. I guess this is important, because it could be possible that the artificial sRNA generated might have a direct effect on expression of genes responsible for the observed phenotype, bypassing the regulation from the ELE RNAs produced by the RLG_famc7.3 TE family.

I am satisfied with all other answers given by authors to my previous questions.

We sincerely thank the reviewers for the positive evaluation and cogent comments to help us improve our manuscript, which we have addressed in detail below.

Reviewer #1 (Remarks to the Author):

In this revised version authors improved their manuscript.

Reviewer #2 (Remarks to the Author):

This is a revised manuscript from Xie and co-workers, in which authors implemented many of my previous comments and produced an improved version, enhancing the text clarity and providing a better explanation for their analyses. There are only two points which remain outstanding, which are:

1) With the new data provided it appears that AG-rich TF binding enrichment is always located upstream of expression on the RLG_famc7.3 family, and this appears to be a structural domain in LTR. This is very interesting, but I was wondering if authors have checked if these AG-rich sequences in the entire wheat genome (not associated to TEs) are associated to generation of ELE RNAs ? In other words, can AG-motifs be considered general promoters for ELE RNAs in wheat, or the association with the TE is important for their expression/function ?

Response: We thank the reviewer for raising the possibility. We calculated the density of AG-rich motifs in the entire genome and TSS regions (± 100 bp) of genes and ELE RNAs. Enhancer like elements (ELEs) were divided to three categories, RLG_famc7.3 derived ELE (RLG_famc7.3-ELE), other types of TE-derived ELE (other-TE-ELE), and ELEs not associated with TEs (non-TE-ELE) (the following figure). These AG-rich motifs are apparently enriched in the TSS region of RLG_famc7.3-ELE. No significant difference was detected between other ELEs and gene TSS regions. Thus, these AG-rich motifs are likely to be associated with specific transcription of RLG_famc7.3-ELE.

Bar plot displays the occurrence of five AG-rich motifs in the TSS region (100 bp up and downstream of TSS) and in the genome background.

2) Figure S19 indeed displays production of sRNAs in knockdown lines. To be informative, I guess an additional track with sRNAseq from a control wild-type non-transformed line should also be presented (showing absence of siRNA accumulation at the same location). Moreover, I guess that in absence of a CAGEseq or RNAseq data performed on these knockdown lines (or by different molecular approach e.g. RT-qPCR), authors cannot assume that an increased amount of siRNA will reflect a real knock-down of the corresponding target Enhancer RNA. I believe in absence of these data, in the manuscript should be mentioned that the suppression of the target, however probable, has not been investigated directly. I guess this is important, because it could be possible that the artificial sRNA generated might have a direct effect on expression of genes responsible for the observed phenotype, bypassing the regulation from the ELE RNAs produced by the RLG_famc7.3 TE family. I am satisfied with all other answers given by authors to my previous questions.

Response: We thank the reviewer for pointing this out. We now included the track of sRNA including both wild-type and RNAi lines in Fig.S19.

We examined the effect of RNAi from two perspectives.

Firstly, the designed sRNA was mapped to sequences of all coding genes, which could be mapped to only three coding genes, all of which have no detectable transcription in spike of wild type wheat. This result suggests that artificial sRNA is not likely directly target and suppress gene expression.

To obtain direct evidence of ELE expression change upon RNA interference, we performed RT-qPCR to detect the RLG_famc7.3 enhancer expression in the knockdown lines. Since ELE-RNAs don't necessarily contain polyA, random primers were used for reverse transcription. We detected significant downregulation of ELE-RNAs in target loci RNAi lines (Fig. S20 and lines 218-220,558-566).

Reviewers' Comments:

Reviewer #2:

Remarks to the Author:

I am satisfied with this version of the manuscript.